# Palmitoylation of the envelope membrane proteins GP5 and M of porcine reproductive and respiratory syndrome virus is essential for virus growth

**Minze Zhang[1], Xiaoliang Han[1,2], Klaus Osterrieder[1,3], Michael Veit[1] ***

**1** Institut für Virologie, Freie Universität Berlin, Berlin, Germany, **2** College of Veterinary Medicine, South China Agricultural University, Guangzhou, China, **3** Department of Infectious Diseases and Public Health, City University of Hong Kong, Kowloon Tong, Hong Kong

* mveit@zedat.fu-berlin.de

**Data Availability Statement:** All relevant data are within the manuscript and its Supporting Information files.

## Abstract

Porcine reproductive and respiratory syndrome virus (PRRSV), an enveloped positive-strand RNA virus in the *Arteiviridae* family, is a major pathogen affecting pigs worldwide. The membrane (glyco)proteins GP5 and M form a disulfide-linked dimer, which is a major component of virions. GP5/M are required for virus budding, which occurs at membranes of the exocytic pathway. Both GP5 and M feature a short ectodomain, three transmembrane regions, and a long cytoplasmic tail, which contains three and two conserved cysteines, respectively, in close proximity to the transmembrane span. We report here that GP5 and M of PRRSV-1 and -2 strains are palmitoylated at the cysteines, regardless of whether the proteins are expressed individually or in PRRSV-infected cells. To completely prevent S-acylation, all cysteines in GP5 and M have to be exchanged. If individual cysteines in GP5 or M were substituted, palmitoylation was reduced, and some cysteines proved more important for efficient palmitoylation than others. Neither infectious virus nor genome-containing particles could be rescued if all three cysteines present in GP5 or both present in M were replaced in a PRRSV-2 strain, indicating that acylation is essential for virus growth. Viruses lacking one or two acylation sites in M or GP5 could be rescued but grew to significantly lower titers. GP5 and M lacking acylation sites form dimers and GP5 acquires Endo-H resistant carbohydrates in the Golgi apparatus suggesting that trafficking of the membrane proteins to budding sites is not disturbed. Likewise, GP5 lacking two acylation sites is efficiently incorporated into virus particles and these viruses exhibit no reduction in cell entry. We speculate that multiple fatty acids attached to GP5 and M in the endoplasmic reticulum are required for clustering of GP5/M dimers at Golgi membranes and constitute an essential prerequisite for virus assembly.

**Funding:** The study was funded by the German Reserch Foundation (DFG), project numbers VE 141/11-2 to MV and OS 143/10-2 to KO. The funders had no role in study design, data collection and analysis, decision to publish, or preparation of the manuscript.

## Author summary

Porcine reproductive and respiratory syndrome virus (PRRSV), an arterivirus in the order *Nidovirales*, is an important pathogen for pigs. Despite its importance in veterinary medicine, basic structural and functional features of its membrane proteins have not been elucidated. Here, we provide evidence for palmitoylation of the PRRSV major membrane proteins GP5 and M at a cluster of membrane-near cysteines. Fatty acid attachment is required for virus growth, since removal of all acylation sites from either M or GP5 prevents recue of infectious particles. Furthermore, viruses lacking individual acylation sites in M and GP5 grow to significantly lower titers in cell culture. The specific infectivity and cell entry of viruses lacking two acylation sites in Gp5 is, however, not reduced. Likewise, these viruses revealed no effect on dimerization of GP5 with M, its transport to budding sites, and incorporation into virus particles. Since cells transfected with a cDNA expressing non-acylated GP5, or non-acylated M release no virus-like particles into the supernatant we propose that the fatty acids are required for the budding process. They might trigger assembly of GP5/M dimers to form a coat inside the lipid bilayer that induces membrane curvature.

## Introduction

Porcine reproductive and respiratory syndrome virus (PRRSV), an enveloped plus-strand RNA virus in the *Arteriviridae* family of the order *Nidovirales*, is one of the most relevant viral pathogens in pigs worldwide. PRRSV infection causes abortion and stillbirth in pregnant sows as well as respiratory disease and poor growth performance in piglets [1]. PRRSV was previously divided into two distinct genotypes referred to as "European" and "North American", but the low sequence identity of only approximately 50% resulted in their re-classification as two distinct species, PRRSV-1 and PRRSV-2, respectively [2]. The early isolates Lelystad virus (LV, PRRSV-1, [3] and VR-2332 (PRRSV-2, [4]) serve as the prototype strains for the two new species. Since their discovery, PRRSV strains have spread worldwide and diversified rapidly by mutation and recombination, resulting in the emergence of highly pathogenic variants in China ([5], related to PRRSV-2) and Eastern Europe ([6], related to PRRSV-1). Besides PRRSV, the *Arteriviridae* family contains other members of veterinary importance, such as the prototype arterivirus, equine arteritis virus (EAV), the murine lactate dehydrogenase-elevating virus (LDV) and simian haemorrhagic fever virus (SHFV) [7, 8].

In Cryo-EM renderings, virus particles appear pleomorphic with an average diameter of 58 nm and a smooth outer surface with only a few protruding features. The virions contain a double-layered, hollow core that contains the viral RNA genome complexed with the nucleocapsid protein N, which is separated from the envelope by a 2–3 nm gap [9]. Arterivirus particles contain a number of membrane proteins, the disulphide-linked GP5/M dimer and the GP2/3/4 complex, the small and hydrophobic E-protein and the ORF5a protein [10]. From reverse genetics experiments, mainly with EAV, it is known that arterivirus structural proteins are essential for virus replication, but act at different steps of the replication cycle. If expression of either Gp2 or Gp3 or Gp4 is abrogated, virus-like particles bud from cells, but the particles are not infectious, indicating that cell entry is disturbed in the absence of the minor glycoprotein complex. In contrast, if either GP5 or M is deleted from the viral genome, no virus particles are released from transfected cells. Thus, GP5 and M are required for virus budding, which does not exclude the possibility that the membrane proteins may have additional functions during virus entry [11, 12].

PRRSV has a restricted cell tropism, infecting primarily porcine macrophages, the natural host cell, and a green monkey kidney epithelial-like cell line (MARC-145, a subclone of MA-104 cells). Initial contacts between the cell and the virus are mediated by binding of M to heparan sulphate proteoglycans, ubiquitous cellular membrane proteins [13]. Sialic acid residues on GP5 are then recognized by sialoadhesins, lectins present on macrophages [14, 15]. Attachment is followed by clathrin-dependent endocytosis of virus particles and their vesicular transport to early endosomes [16, 17]. In the endosome, the Gp2/3/4 complex binds to CD163, a scavenger receptor for hemoglobin clearance [18, 19]. CD163 is essential for virus entry, since pigs with deletions in this gene are not susceptible to PRRSV infection [20–22]. After receptor-mediated endocytosis of virus particles, a drop in endosomal pH, probably accompanied by the proteolytic activity of cathepsin E [23], activates the membrane fusion machinery of PRRSV. However, it has not been identified so far for any of the arteriviruses which of the viral spikes mediates membrane fusion.

GP5 (~200 amino acids in length) is the major glycoprotein of PRRSV and a target for neutralizing antibodies [24]. It comprises an N-terminally cleaved signal peptide directing protein synthesis at the rough endoplasmic reticulum (ER), followed by an ectodomain of roughly 30 amino acids. It contains two highly conserved N-glycosylation sites, and, depending on the virus strain, no, one or two additional carbohydrates are attached to a hypervariable region adjacent to the signal peptide. The region between residues 65 and 127 is hydrophobic and assumed to span the membrane three times. The C-terminal part (~75) is hydrophilic and located in the cytosol, ending up in the virus interior [25]. GP5 of PRRSV is the most variable structural protein with only ~50% amino acid homology between PRRSV-1 and PRRSV-2 strains [26, 27].

M (~174 amino acids, predicted MW ~18 kDa) is the most conserved membrane protein of the arteriviruses. It consists of a short ectodomain (17 amino acids), three putative transmembrane regions (20 amino acids each), which are connected by two short loops, and a long, hydrophilic cytoplasmic tail (77 amino acids). The short N-terminal ectodomain does not contain a signal peptide or a consensus sequence for the attachment of N-linked glycans [25].

GP5 forms a disulphide-linked complex with M involving the only cysteines in their extracellular domains, cysteine (Cys) 50 in GP5 and Cys 9 in M [10]. For equine arteritis virus (EAV) it was shown that heterodimerization of M with GP5 is required for their transport from the ER to the Golgi apparatus where they are retained by unidentified signals [28]. Except for the pivotal role of the GP5/M dimer, the molecular mechanism of arterivirus budding is largely unexplored. According to electron microscopy studies, virus particle generation proceeds by budding of pre-formed nucleocapsids into the lumen of organelles of the exocytic pathway, presumably the Golgi apparatus. After budding of the virions, they are transported along the secretory route and ultimately released by exocytosis [29].

S-acylation (or palmitoylation) is the post-translational attachment of fatty acids to Cys residues of membrane proteins [30]. Palmitoylation was originally discovered for the G protein of Vesicular Stomatitis Virus (VSV) [31]. It rapidly became evident that (almost) every enveloped virus contains an S-acylated type 1 transmembrane protein, often with membrane fusion activity. Another category of palmitoylated viral proteins are viroporins, small hydrophobic proteins with one or two membrane-spanning regions that often function as ion channel during virus entry and/or organize virus assembly and scission of particles. The third category of palmitoylated viral proteins are intrinsically hydrophilic polypeptides that are attached to membranes by fatty acid attachment [32].

It was reported for influenza virus hemagglutinin (HA), the spike protein S and the viroporin E of coronaviruses that the modification is beneficial or even essential for virus replication. Fatty acids play a role during virus entry by membrane fusion and/or during assembly and budding of new virus particles [33–37]. In these cases, the palmitoylated cysteines are conserved through all

virus strains, which possess otherwise highly variable glycoproteins. In contrast, no function could be attributed to palmitoylation of other viral membrane proteins, such as G of VSV and M2 of Influenza A virus, and virus strains exists that do not contain a palmitoylated cysteine [38, 39]. In transmembrane proteins, palmitoylated cysteines are located at the cytosolic site of the transmembrane region and in the cytoplasmic tail, often close to the membrane-spanning domain. In contrast to most other protein modifications, no consensus motif that directs palmitoylation has been established and current prediction tools, such as Css Palm 4 (http://csspalm. biocuckoo.org/online.php) are of limited value to predict palmitoylation of viral proteins [40].

Since both M and GP5 of PRRSV possess a cluster of highly conserved cysteine residues in the cytoplasmic tail next to the third transmembrane region (Fig 1) we examined here whether the cysteines are palmitoylated and whether the fatty acid modification is required for virus replication.

## Results

### GP5 and M of PRRSV-1 and PRRSV-2 are palmitoylated in infected and transfected cells

To identify conserved cysteine residues as putative palmitoylation sites in M and GP5, 1077 ORF5 nucleotide sequences encoding GP5 from PRRSV-1 strains, 7701 from PRRSV-2 strains, 83 ORF6 nucleotide sequences encoding M from PRRSV-1 strains and 290 from PRRSV-2 strains present in the NCBI database were translated into the corresponding amino acid sequences. From these data a consensus sequence for GP5 and M of both PRRSV-1 and PRRSV-2 containing the most abundant amino acid at each position was compiled, and the percent conservation at each position was plotted against the amino acid number. The results reveal that three cysteines in GP5 are 100% conserved in each virus species, one at the end of the transmembrane region (TMR) and two in the cytoplasmic tail (Fig 1A). The position of the first two cysteines is also conserved between PRRSV-1 and PRRSV-2 strains, but the last cysteine is shifted by four residues towards the C-terminus of GP5 in the PRRSV-2 strains. M also contains cysteines at the membrane-proximal side of the cytoplasmic tail, which are completely conserved in each virus species, three in M of PRRSV-1 strains and two in M of the PRRSV-2 strains. Likewise, alignment of GP5 and M amino acid sequences of representative members of other Arterivirus genera, such as equine arteritis virus (EAV), lactate dehydrogenase elevating virus (LDV) and simian haemorrhagic fever virus (SHFV) revealed that all of them, except M of EAV, have cysteines located in the membrane-proximal part of the cytoplasmic tail (Fig 1B).

To investigate whether GP5 and/or M are palmitoylated, we infected MARC-145 cells with PRRSV-2 strains VR-2332 or XH-GD at a low multiplicity of infection (MOI). Forty-eight hours post-infection (p.i.) cells were lysed and subjected to Acyl-RAC (resin-assisted capture), which exploits thiol-reactive resins to capture SH-groups in proteins. Approximately 10% of the total extract (TE) was removed from the lysate to determine the expression levels of GP5. Disulphide-bonds in proteins present the remaining part were reduced and newly exposed -SH groups blocked. The sample was then equally split: one aliquot was treated with hydroxylamine to cleave thioester bonds, the other aliquot was treated as a control with Tris-Cl buffer. After pull-down of proteins with the thiol-reactive resin, samples were subjected to western blotting using monoclonal antibodies against GP5 and M of PRRSV-2 strains (Fig 2A). A clear signal was present for M (sharp band with a molecular weight of 19kDa) and GP5 (diffuse bands between 23-35kDa) in hydroxylamine treated (+HA), but not in Tris-treated samples (-HA) indicating that both proteins are acylated. Palmitoylation of M and GP5 was also detected during replication of both viruses in isolated porcine alveolar macrophages, the target cells of PRRSV in pigs (Fig 2B).

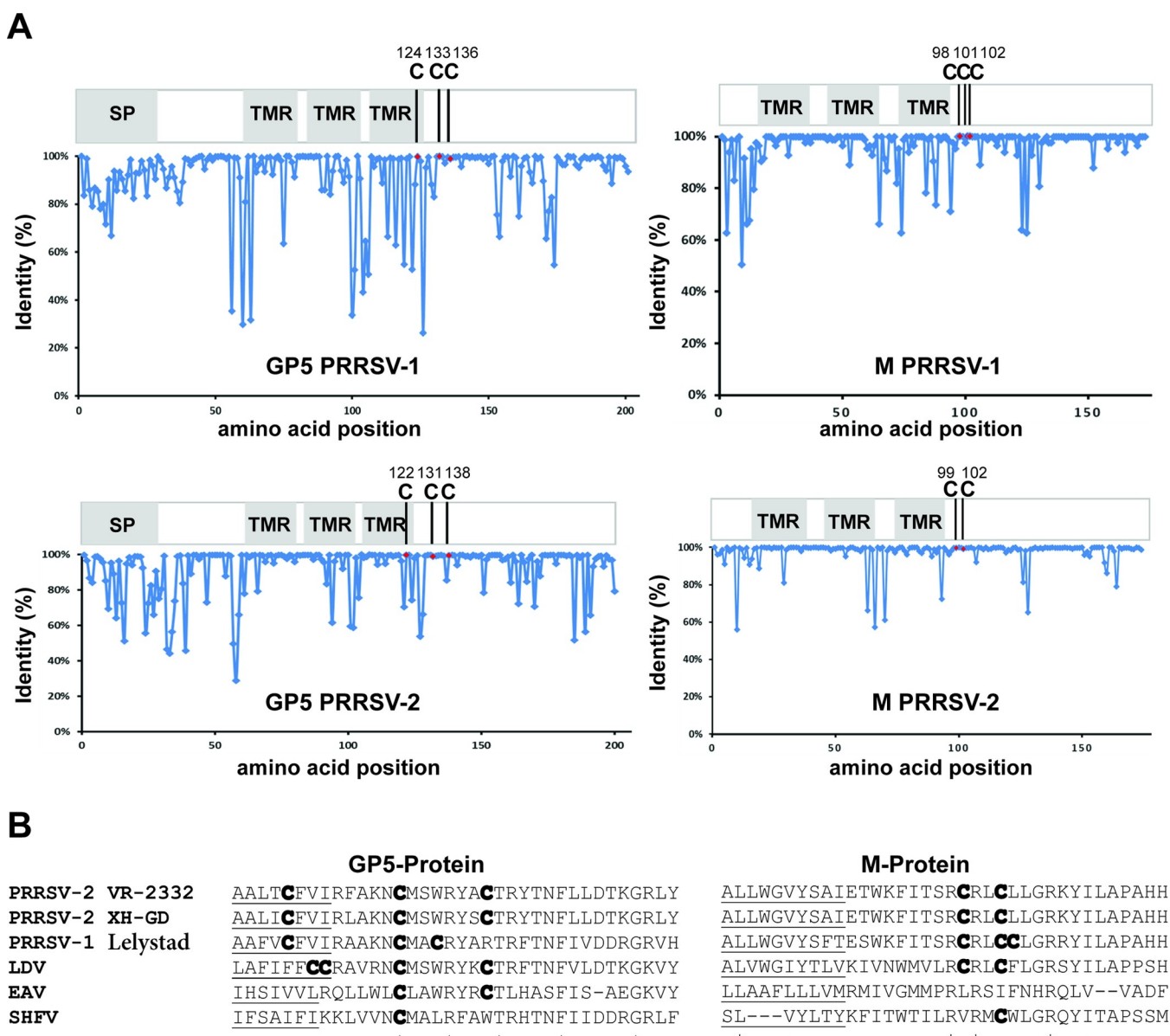

**Fig 1. Conservation of cysteine residues in the cytoplasmic tail of GP5 and M.** (A) The graphs show the percent conservation (y axis) of amino acids at each position (x axis) of a consensus sequence compiled from all M and GP5 sequences of either PRRSV-1 and PRRSV-2 present in the database. The scheme above each graph indicates the location of the signal peptide (SP) and transmembrane region (TMR) within the primary structure of the respective protein. C are the conserved cysteine residues and the number indicates their position in the amino acid sequence. (B): Aligned amino acids sequences of the transmembrane region (TMR, underlined) and the adjacent part of the cytoplasmic tail of GP5 and M of different members of the Arteriviruses. VR-2332: type 2 prototype PRRSV strain, XH-GD: highly pathogenic Chinese PRRSV-2 strain, Lelystad: type 1 prototype PRRSV strain, LDV: lactate dehydrogenase-elevating virus, EAV: equine arteritis virus prototype strain Bucyrus and SHFV: simian haemorrhagic fever virus. Cysteines are marked in bold. Asterisk (*): Identical amino acids, colon (:): residues with strongly and period (.): with weakly similar properties.

To investigate whether M and GP5 of the PRRSV-1 prototype strain Lelystad, for which no antibodies were available to us, are also acylated, we added to their C-termini either a His-tag (M) or an HA-tag (GP5). These constructs as well as equivalent constructs of the VR-2332 strain were transfected into 293T cells, which were subjected to Acyl-Rac that revealed a strong signal in hydroxylamine-treated samples for GP5-HA and M-His of both viruses (Fig 2C and 2D). It is worth mentioning that GP5 in transfected cells was detected as one band after

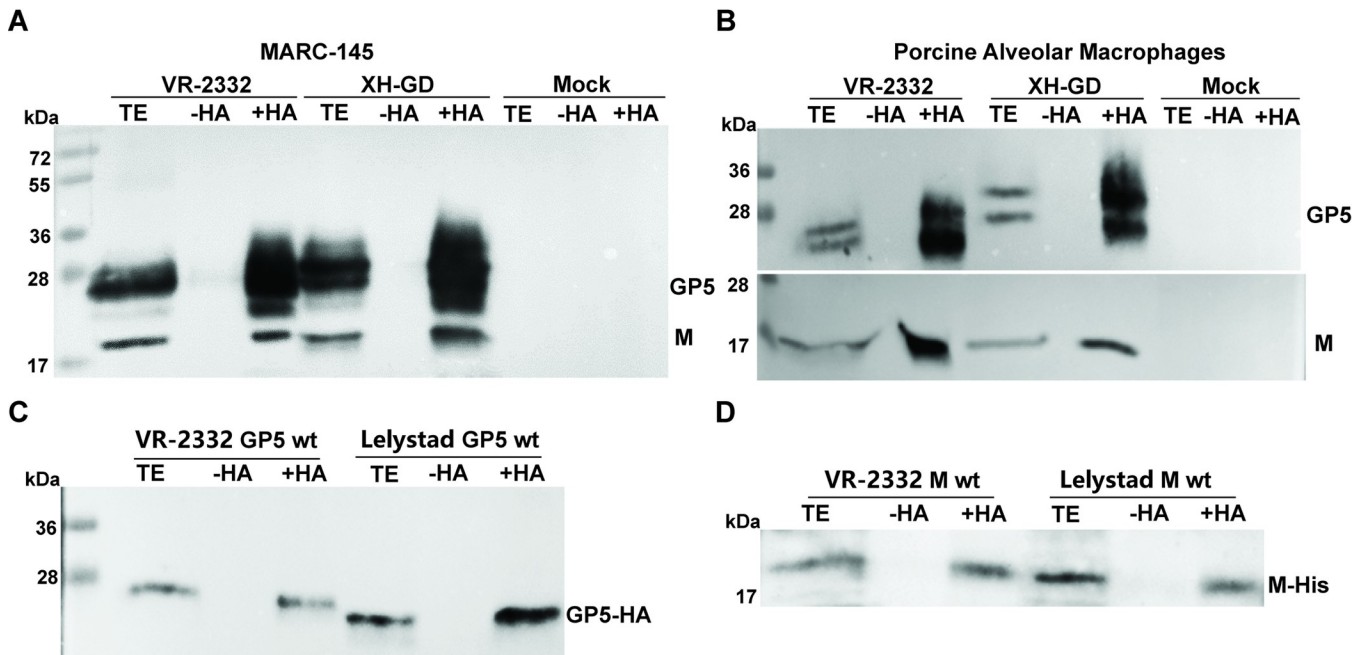

**Fig 2. Palmitoylation of GP5 and M of PRRVS-1 and PRRSV-2 strains.** (A) GP5 and M of two PRRSV-2 strains are palmitoylated in infected MARC-145 cells. Cells were left uninfected (mock) or infected with VR-2332 or XH-GD at MOI of 0.001, lysed after 48h and subjected to Acyl-Rac to detect palmitoylated proteins. Cell lysates were divided into two aliquots either treated (+HA) or not treated with hydroxylamine (-HA). TE: total extraction: 10% of the cell lysate before hydroxylamine treatment. Samples were analyzed by western blotting with a mixture of monoclonal antibodies against GP5 and M. (B) GP5 and M of two PRRSV-2 strains are palmitoylated in infected PAM cells. Cells were left uninfected (mock) or infected with VR-2332 or XH-GD at MOI of 0.001, lysed after 48h and subjected to Acyl-Rac to detect palmitoylated proteins. Palmitoylation was analyzed with Acl-Rac as described in legend to Fig 1A except that monoclonal antibodies against GP5 and M were applied to different membranes. (C) and (D) Palmitoylation of GP5-HA and M-His in transfected 293T cells. GP5 (equipped with a C-terminal HA-tag) and M (equipped with a C-terminal His-tag) from VR-2332 and Lelystad strains were expressed in 293T cell. 48 hours after transfection acylation of GP5 and M was analyzed using Acyl-RAC and Western Blot with antibodies against the HA-tag (C) or the His-tag (D). The left part of each blot shows the mobility of molecular weight markers with the indicated masses in kDa.

SDS-PAGE, in contrast to the diffuse bands of GP5 in virus-infected cells, which is likely a result of heterogenous processing of N-glycosylation sites in the Golgi [41].

The results suggest that GP5 and M remain in the ER if expressed alone, consistent with previous results [12] and as also reported for M and GP5 of Equine Arteritis Virus (EAV) [28]. Acylation of other viral glycoproteins, such as hemagglutinin of Influenza virus occurs in an early Golgi-region by enzymes located in the same compartment [42, 43]. To test how much of GP5 and M is transported to the Golgi, we analysed co-localization of GP5-HA and M-His of the VR-2332 strain with the cis-Golgi-marker GM 130 in transfected BHK cells by confocal microscopy. Both proteins exhibited a mainly reticular staining pattern that is typical for the ER. Only limited overlap in the perinuclear region with the cis-Golgi marker is visible (S1 Fig). Quantification of 40 cells revealed that ~12% of M-His and ~8% of GP5-HA molecules co-localize with GM 130. We conclude that monomers of GP5 and M are the substrate for acylation, i. e. dimerization of GP5 with M is not required. Furthermore, both GP5 and M are probably acylated by enzymes localized in the ER, since only a subfraction of either protein was present in the Golgi apparatus.

## Fatty acids are attached to all putative acylation sites in M and GP5

To determine whether the candidate cysteines indeed serve as acylation sites (Fig 1A), we exchanged them with serine in GP5-HA and M-His from both the VR-2332 and Lelystad

viruses. The resulting constructs, named GP5 SSS and M SS, were expressed in 293T cells. Acyl-Rac showed that for any mutant protein palmitoylation was no longer detectable (Fig 3A–3D). To exclude that proteins were lost during sample preparation, we used antibodies against the cellular palmitoylated protein flotillin 2 as an internal control.

Whether all three cysteines are acylated in GP5 of VR-2332 and Lelystad was tested by changing them to serine individually or in all possible dual combinations. Exchange of any single cysteine did not completely prevent, but reduce, palmitoylation of GP5 of both virus strains (Fig 3E and 3G, compare TE and +HA-signals of GP5-HA with those of the mutants). However, not every cysteine had the same effect on the extent of acylation. Substitution of the central cysteine (Cys 133 in Lelystad and Cys 131 in VR-2332) caused the strongest reduction in palmitoylation of GP5 of both strains. Quantification of the band densities from this and two other experiments revealed a remaining palmitoylation level of 29% in GP5 from the Lelystad and of only 17% in GP5 from the VR-2332 virus. If this cysteine was exchanged together with the C-terminal cysteine 136 (Lelystad) or cysteine 138 (VR-2332), palmitoylation of GP5 was virtually undetectable (Fig 3F and 3H). The C-terminal cysteines, if exchanged alone, had a very modest effect on the palmitoylation levels. Total palmitoylation was insignificantly reduced to ~55% (Lelystad) or 90% (VR-2332), whereas substitution of the transmembrane cysteine reduced fatty acid attachment to ~50–60% in GP5 of both virus strains. We concluded from our experiments that all three cysteines in GP5 are acylated, but especially the central cysteine is most important for efficient palmitoylation.

## Palmitoylation of both M and GP5 is essential for virus growth

In the next series of experiments, we analysed whether palmitoylation of GP5 and M affects virus replication. We used a reverse genetics system based on a full-length cDNA clone of the Chinese PRRSV-2 strain XH-GD cloned into a plasmid downstream of the cytomegalovirus (CMV) immediate-early (IE) promotor. First, we created mutants where all three cysteines in GP5 or both in M were exchanged to serine and another construct, which contains both mutant genes in one viral genome. Full-length cDNA was transfected into BHK cells, which lack the receptors for cell entry of PRRSV, but can release infectious virus particles, and 48 hours after transfection the cell culture supernatant was used to infect MARC-145 cells, which support virus replication. Cells transfected with wild-type or mutant cDNA express viral proteins, as shown by immunofluorescence with antibodies against GP5 (S2A Fig). However, infected cells showed a fluorescence signal only for wild-type virus, but not for any of the three mutants. Furthermore, using qRT-PCR, we could not detect viral genomes for any of the GP5 and M mutants in the supernatant of infected cells, whereas ~$10^7$ genome copies of wild-type virus were present. Likewise, five further attempts to rescue infectious mutant virus, in two of them we harvested the cell culture supernatant 72 or 96 hours after transfection, also failed, but wild-type PRRSV was rescued in each case. We concluded from our experiments that PRRSV acylation mutants synthesize GP5 in transfected cells, but cells apparently do not release infectious virus particles.

The next series of viral mutants that we engineered encode GP5 proteins in which the three cysteines were exchanged individually or in all possible combinations to serine. In addition, a mutant was created in which the two cysteines of M were individually exchanged. Immunofluorescence of transfected cells revealed that all mutant viruses expressed GP5 and produced virus particles that can infect MARC-145 cells (S2B Fig). The exception was the double mutant GP5 131+138, which, however, is not a target of acylation in the closely related VR-2332 strain and thus has the same phenotype as GP5 SSS, which lacks all acylation sites.

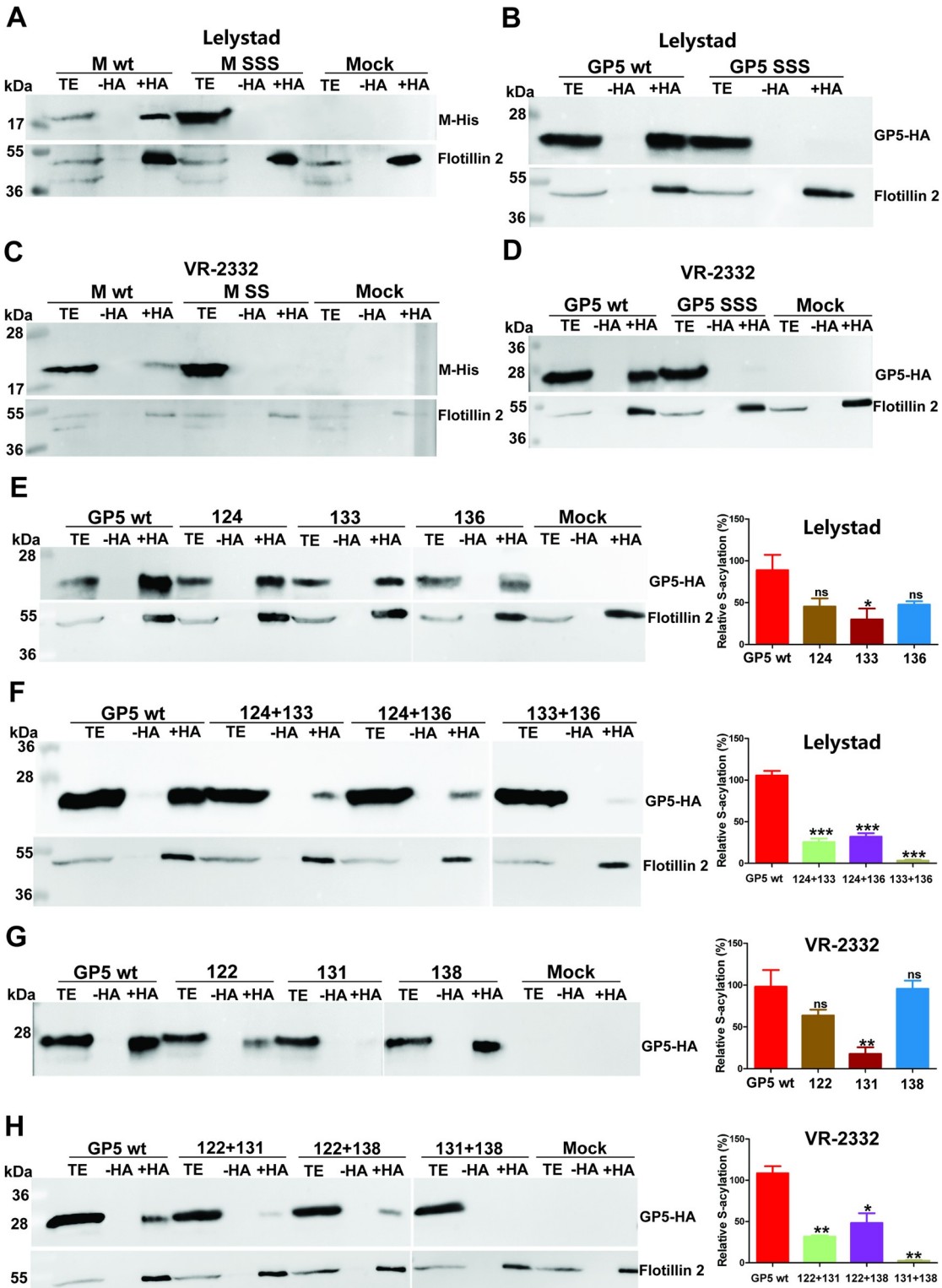

**Fig 3. Determination of palmitoylation sites in GP5 and M of PRRSV-1 and PRRV-2 strains. (A-D):** All putative palmitoylation sites in M (A+C) and GP5 (B+D) of PRRSV-1 prototype strain Lelystad (A+B) and PRRSV-2 prototype strain VR-2332 (C+D) were exchanged to serine. Plasmids with the indicated mutations were transfected into 293T cells and palmitoylation was determined with Acyl-Rac. Anti-His antibodies were used to detect M-His and anti-HA antibodies to detect GP5-HA. Staining of the same membrane with antibodies against the cellular palmitoylated protein Flotillin 2 served as a control for sample processing. **(E-H):**

Cysteines were exchanged individually (E+G) or in all possible double combinations (F +H) in GP5 of Lelystad (E+F) and VR-2332 (G+H). Plasmids with the indicated mutations were transfected into 293T cells and mutants analyzed for palmitoylation. The right panel of each figure shows quantification of the results from this and two other identical experiments. The optical densities of the +HA (palmitoylation) and TE (expression level) bands were determined, the ratios were calculated and normalized to the mean value of GP5-HA wild type (= 100%). The mean ± standard deviation is shown. The asterisks indicate statistically significant differences (*P < 0.05, **P < 0.01, ***P < 0.001) between GP5-HA wild type and the GP5 cysteine-mutant. One-way ANOVA followed by Tukey's multiple comparison test was applied for statistical analysis.

## Palmitoylation of GP5 and M affects release of virus particles

For both PRRSV and EAV it was reported that blockade of either GP5 or M expression in an infectious clone prevents not only rescue of infectious virus, but release of virus-like particles [11, 12]. To analyse whether cells transfected with a cDNA expressing non-acylated GP5, or non-acylated M or both mutant proteins together exhibit the same phenotype, we used qRT-PCR to estimate the number of viral genomes released into the supernatant of transfected BHK cells. As a negative control we created a mutant where expression of GP5 was completely abrogated by elimination of the start codon. We also tested the mutants GP5 131+138, which expresses non-acylated GP5 and GP5 122+131, where acylation of GP5 is reduced, but infectious virus could be rescued. BHK cells were transfected with the plasmids, the supernatant was removed 48 hours later and treated with benzonase to remove any remaining plasmid DNA. An aliquot of the supernatant and also of a cellular lysate was reverse transcribed into DNA, a fragment of the Gp3 gene was amplified and the gene copy number estimated by qRT-PCR. Around 1000 gene copies (mean of three transfections) were detected in the supernatant of $1 \times 10^6$ cells transfected with wt cDNA and ~800 gene copies for the cDNA encoding the mutant GP5 122+131 (Fig 4). However, not any viral genomes were found for the other cDNAs in the three transfection experiments, although the number of intracellular gene copy numbers and Gp3 messenger RNAs are the same in cells transfected with the different constructs. The results confirm that palmitoylation of both GP5 and M is essential for production of infectious virus particles and suggest that the release of particles might be the underlying defect.

## PRRSV with underacylated GP5 and M grow to lower titers

To analyse the replication kinetics of the rescued viruses, MARC-145 cells were infected with wild-type or one of the mutant viruses at an MOI of 0.01, supernatants were collected at various times, and titers were assessed by $TCID_{50}$ assay. The results from three independent infections revealed that each mutant virus grows to lower titers when compared to those of wild-type PRRSV. If two cysteine were exchanged in GP5, virus titers were reduced by up to 10,000-fold at 48, 72 and 96 hours post infection (p.i.) (Fig 5B). If cysteines were exchanged individually, the reduction in titers was less pronounced and reached between 10- to 100-fold at 48 hours p.i., the time when viral titers peaked. Titers were also reduced at earlier and later time points, but these results were not significantly different from those recorded for wild-type virus (Fig 5A). It is also worth noting that all growth curves exhibited the same shape, indicating that a reduction of viral titers was not due to a delay in the onset of virus replication.

Likewise, removing the two cysteines individually in M caused a statistically significant titer reduction by approximately 10-fold (M 99) or 1,000-fold (M 102) at 48 hours p.i., and a tendency to lower titers at other time points (Fig 5C). We also used the engineered viruses to show by Acyl-Rac that both cysteines in M are acylated in virus-infected cells (Fig 5D). Exchange of Cys 99 caused a reduction in palmitoylation to ~20% and of Cys 102 to ~35% relative to wild-type GP5 (set to 100%, mean of three experiments).

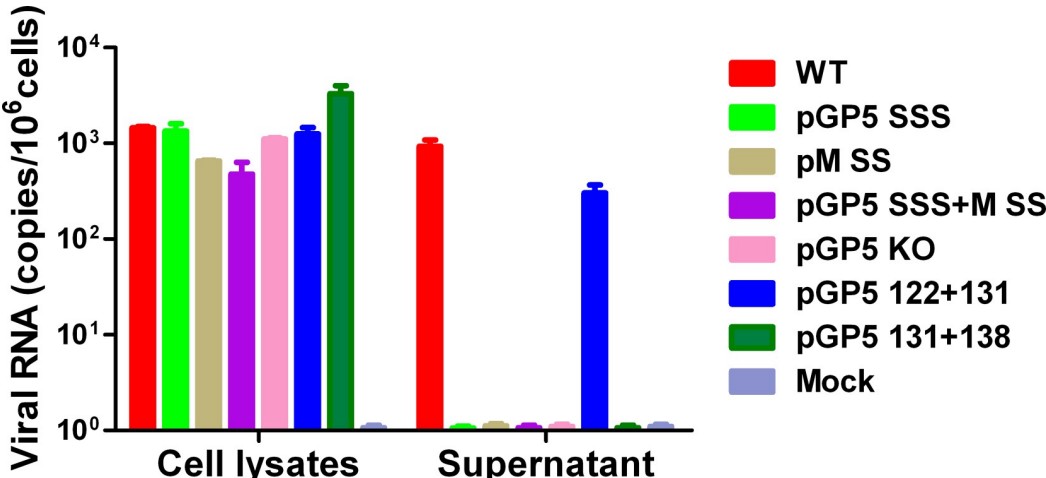

**Fig 4. Palmitoylation of GP5 and M is required for release of virus particles from transfected cells.** Quantification of viral genomes present in cellular lysates or released into the supernatant of BHK cells transfected with full-length PRRSV plasmids as indicated. GP5 KO is a mutant where expression of GP5 was abrogated by mutation of its start codon. 24 hours post-transfection, viral RNA was isolated from cleared and benzonase-treated culture supernatants and from cell lysates and reverse transcribed into cDNA. Q-PCR was done with primers that amplify a fragment of the viral GP3 gene. Full-length plasmid of an infectious wild-type clone was as used as a standard to estimate the number of gene copies. Results from three independent transfections are shown as number of viral RNA (mean including standard deviation) present in or released into the supernatant of $1 \times 10^6$ transfected cells.

## Palmitoylation is not required for dimerization of GP5 and M, for transport to the Golgi and for incorporation into virus particles

It was reported for both EAV and PRRSV that GP5 and M are the forces driving virus budding [11, 12]. As far as GP5 and M are concerned, virus budding can be divided into the following steps: formation of a disulfide-linked dimer, its transport out of the ER to the Golgi, assembly of GP5/M dimers at the budding site, and their incorporation into virus particles. We determined whether any of these steps is compromised in mutant viruses. We first tested whether deletion of palmitoylation sites influences disulphide-induced dimerization of GP5 with M. This is of particular interest since arteriviruses lacking the cysteine mediating dimerization exhibit the same phenotype as the palmitoylation mutants, i.e. they are unable to produce infectious virus particles [28]. We infected MARC-145 cells with wild-type and acylation-deficient mutants, separated cell lysates by SDS-PAGE under reducing or non-reducing conditions, and visualized GP5 by western blotting. Under reducing conditions (+mercaptoethanol, ME) GP5 was detected as two diffuse bands with molecular weights of 23kDa and 27kDa, respectively. Under non-reducing SDS-PAGE (-ME) an additional, disulphide-linked band of higher molecular weight (~38 kDa) appeared. Quantification of the dimerization ratio from this and two other infections and normalization of the value to PRRSV wild-type virus revealed slightly higher, but insignificantly different values for every acylation deficient mutant (Fig 6A).

S-acylation of two transmembrane proteins, the yeast chitin synthase Chs3 and the Wnt signaling co-receptor LRP6, was shown to be required for ER export [44, 45]. To investigate whether palmitoylation of GP5 or M facilitates transport to the Golgi, we treated cell lysates with PNGase F, which cleaves all types of N-linked carbohydrates, or with endo-beta-N-acetyl-glucosaminidase (Endo H), which cleaves only high-mannose-type carbohydrates typical for ER-localized proteins. After digestion of GP5 with PNGase F, only one band with a molecular weight (MW) of 17 kDa was present, which suggests that the multiple bands of untreated GP5 are due to heterogenous N-glycosylation. Endo-H digestion also produced the 17 kDa band,

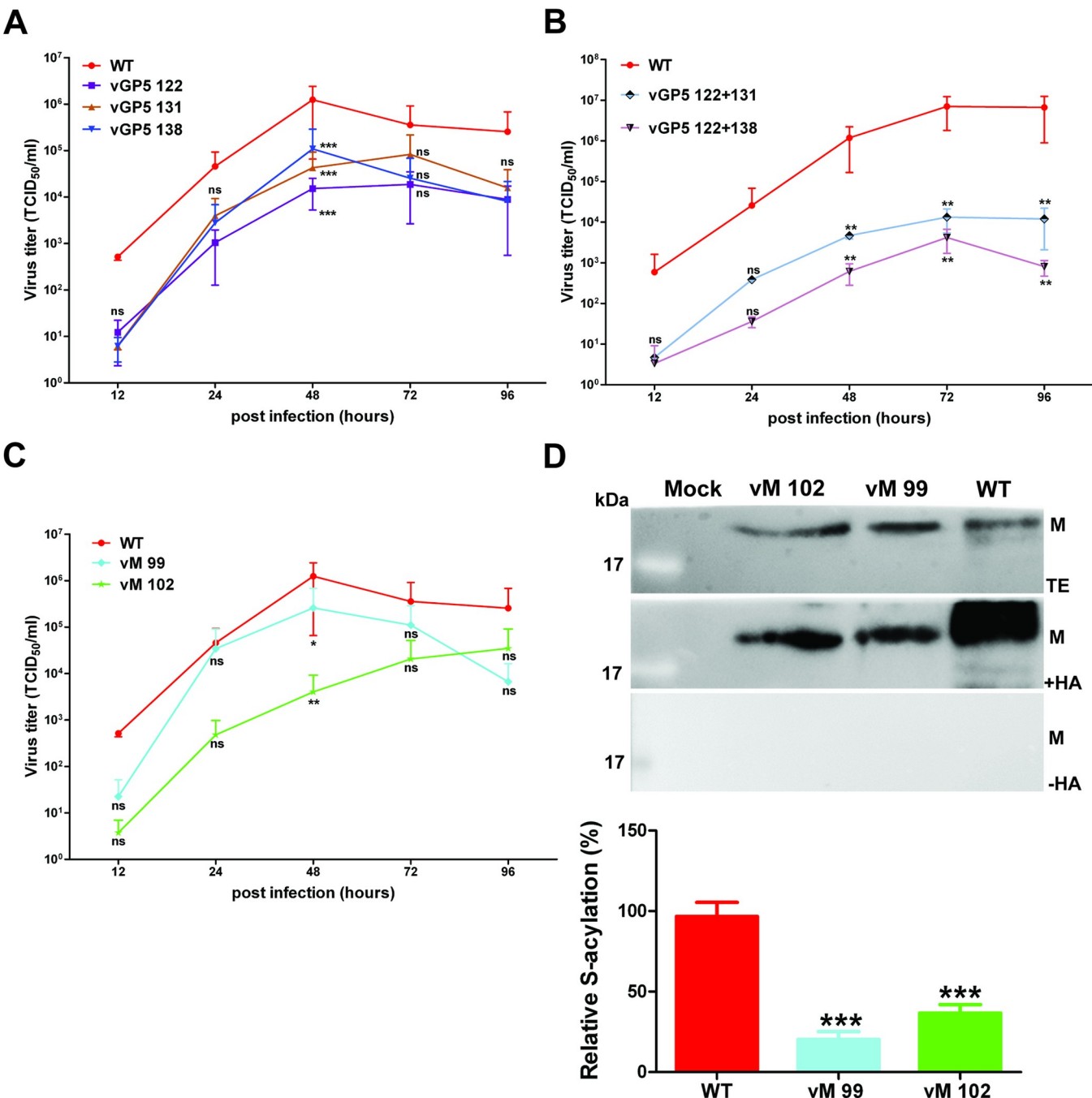

**Fig 5. Growth kinetics of PRRSV acylation-mutants.** (A-C) Growth kinetics of WT and PRRSV cysteine-mutants of M and GP5. (A) PRRSV wt and mutants where each cysteine in GP5 was exchanged to serine. (B) PRRSV wt and mutants where two cysteines in GP5 were exchanged to serine. (C) PRRSV wt and mutants where each cysteine in M was exchanged to serine. Cells in 24-well plates were infected with PRRSV at a MOI of 0.001. Culture supernatants were collected at the indicated time points and titers were determined with a TCID 50 assay. The geometric mean titers with standard deviations (error bars) from two or three independent infections are shown. The asterisks indicate statistically significant differences (*P < 0.05, **P < 0.01, ***P < 0.001) at the same time point between WT and mutants. Two-way ANOVA followed by Bonferroni post-test was applied for comparing replicate means. (D) Analysis of palmitoylation of M. MARC-145 cell were infected with WT or M mutants at MOI of 0.001. Acylation was analyzed after 48h using Acyl-RAC and Western blot with antibodies against M. The bar chart shows the quantification (mean ± standard deviation) of the result from this and two other identical infections. The asterisks indicate statistically significant differences (*P < 0.05, **P < 0.01, ***P < 0.001) between WT and the mutants. ns: no statistically significant difference. One-way ANOVA followed by Tukey's multiple comparison test was applied for statistical analysis.

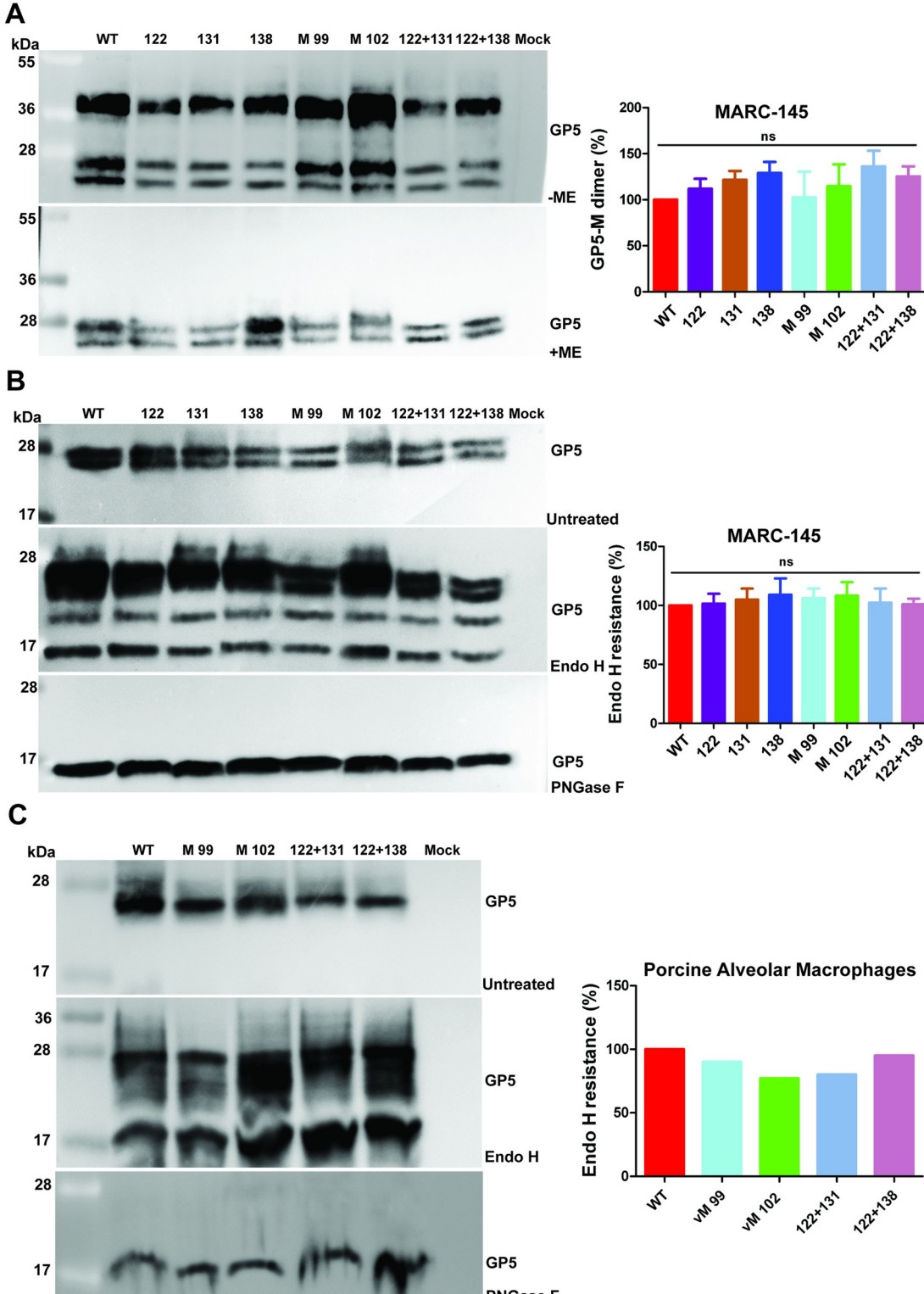

**Fig 6. Palmitoylation of GP5 and M is not required for dimer formation and its transport to the Golgi-complex.** (A) Formation of the disulfide-linked GP5-M dimer. MARC-145 cells infected with WT or mutant viruses at an MOI of 0.001 were lysed at 48 h p.i. and separated by non-reducing (ME) or reducing SDS-PAGE (+ME) followed by western blot using GP5 antibodies. The band visible only under non-reducing conditions and running with the molecular weight of 38kDa is the GP5 M dimer, since GP5 does not form homodimers [49]. The optical density of bands in the samples separated under non-reducing

conditions of this and two other experiments was determined and the percentage of dimer formation was calculated as dimer divided by the sum of all three bands and normalized to PRRSV wt. (B) Endo H resistance of GP5. MARC-145 cells infected with WT or mutants were lysed at 48 h p.i. and either treated with Endo H (middle panel), with PNGase F (lower panel) or left untreated (upper panel) prior to reducing SDS-PAGE and Western blot with GP5 antibodies. The optical density of bands of this and two other experiments was determined and the percentage of Endo H resistant bands was calculated as first and second band (= Endo H resistant bands) divided by the sum of all three bands and normalized to PRRSV wt. (C) Endo H resistance of GP5. PAMs infected with WT or mutants were lysed at 48 h p.i. and either treated with Endo H (middle panel), with PNGase F (lower panel) or left untreated (upper panel) prior to reducing SDS-PAGE and Western blot with GP5 antibodies. The optical density of bands from this experiment was determined and the percentage of Endo H resistant bands was calculated as described in B. The right panel of each figure presents quantification of the result. The mean ± standard deviation is shown for A and B. One-way ANOVA followed by Tukey's multiple comparison test revealed no significant (ns) differences between wt and mutant proteins.

but an additional band with a MW of around 25 kDa. The latter likely represents the small number of GP5 molecules where only two of the three carbohydrates acquired Endo-H resistance. However, the overwhelming majority of GP5 molecules is resistant to enzymatic digestion (Fig 6B).

The observed pattern after enzymatic deglygosylation is consistent with mass spectrometric analysis of carbohydrates attached to GP5 of purified virus particles, which revealed a large variety of different carbohydrates, mostly complex-type N-glycans, but also a small fraction of Endo-H sensitive sugars [41]. Quantification of the ratio of Endo-H resistant GP5 from this and two other infections and normalization of the values to PRRSV wild-type virus revealed no significant difference and thus the same number of molecules have passed through the medial Golgi-complex, where glycoproteins acquire Endo-H resistance [46].

We repeated the same experiment with porcine alveolar macrophages (PAMs) infected with wild-type PRRSV and the mutants lacking two acylation sites in GP5 or the single acylation sites in M. The results show that the majority of GP5 molecules become Endo-H resistant, regardless of the acylation status of GP5 or M (Fig 6C).

We also investigated trafficking of non-acylated GP5-HA and non-acylated M-His of the VR-2332 strain in transfected 293T cells. The carbohydrates attached to GP5, either acylated GP5 or GP5 with the three acylation sites exchanged remained completely Endo-H sensitive if the proteins were expressed alone confirming that the majority of molecules remain in the ER (S3 Fig). Co-expression of both wild-type proteins revealed Endo-H resistant GP5, but the majority (~80%) of molecules remained Endo-H sensitive similar to what we previously reported for the PRRSV-1 strain Lelystad [47]. Nevertheless, Endo-H resistant bands were also detected if non-acylated GP5-SSS was co-expressed with either wildtype M or with M SS lacking the two acylation sites.

Confocal microscopy on transfected BHK cells using antibodies against the HA-tag and the His-tag revealed that wild-type GP5 colocalizes with wild-type M in the perinuclear region, most likely the Golgi (Fig 7, compare with S1 Fig). The same co-localization pattern was seen if wild-type GP5 was co-expressed with M-SS, wild-type M with GP5-SSS and GP5-SSS with M-SS. Quantification of the degree of co-localization of the two fluorophores using Pearson's correlation revealed that ~90% of GP5 wild-type molecules co-localize with wild-type M and the same values were obtained for non-acylated variants of both proteins. Since oligomerisation of glycoproteins is one prerequisite for their export from the ER [48], it is reasonable to assume that complete removal of acylations sites from GP5 and M does not affect their dimerization. We concluded that processing of GP5, dimer-formation with M, and subsequent transport to the Golgi are not affected by removal of palmitoylation sites. Since a large fraction of GP5 remains Endo-H sensitive upon co-expression with M, we assume that the perinuclear region where M and GP5 co-localize represents the cis-Golgi. However, more work is required to precisely identify this organelle, which likely represents the viral budding site.

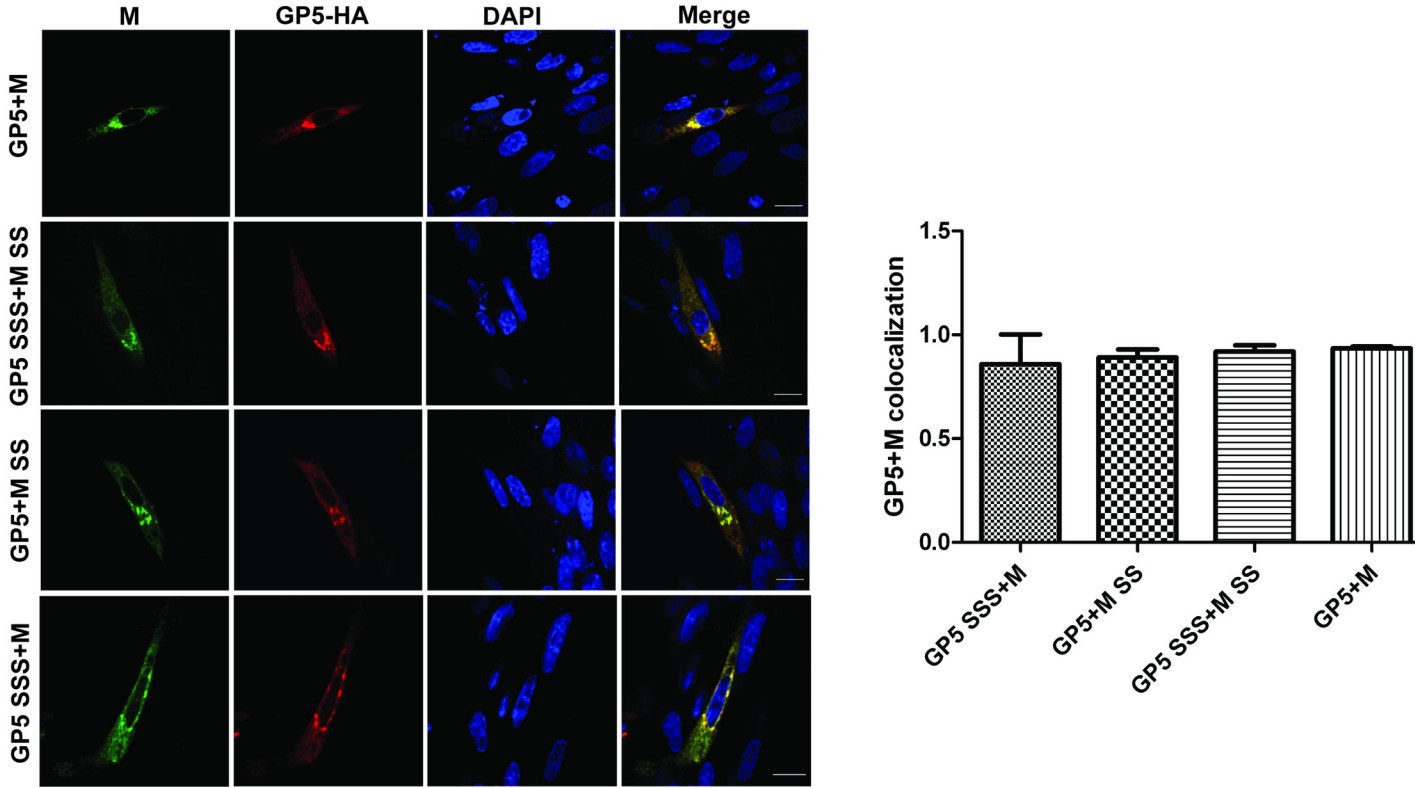

**Fig 7. Non palmitoylated GP5 co-localizes with M in a perinuclear region.** BHK-21 cells were transfected with GP5-HA wt and M-His wt (upper panel), or with non-acylated GP5-HA SSS and M-His SS (second from above), or with GP5-HA wt and non-acylated M-His SS (second from below), or with non-acylated GP5-HA SSS and M-His wt (lower panel). GP5 was stained with polyclonal antibodies against the HA-tag followed by secondary anti-rabbit antibodies coupled to Alexa Fluor-568 fluor, M with monoclonal anti-M antibodies followed by secondary anti-mice antibodies coupled to Alexa Fluor-488 and nuclei with DAPI. Right panel: Co-localization of M with GP5 from at least 40 cells was quantified with the Pearson's correlation coefficient method using the JACoP plugin of the ImageJ software. Around 80% of M and GP5 pixels overlapped regardless of whether they are palmitoylated or not.

Next, we asked whether palmitoylation of GP5 facilitates incorporation of the protein into virus particles. Wild-type PRRSV particles and particles from the GP5 mutants with two exchanged cysteines were prepared from the supernatant of infected MARC-145 cells by centrifugation through a sucrose cushion and the amount of GP5 relative to the nucleocapsid protein N was assessed by western blotting (Fig 8). Quantification of band intensities from three virus preparations revealed a small reduction of the normalized GP5/N ratio for the mutants (GP5 122+131 to ~60%, GP5 122+138 to ~70%), which, however, was not significantly different from wild-type PRRSV (= 100%). Since at least for EAV it was demonstrated that only GP5/M heterodimers are incorporated into virions we assume that the level of M in virus particles is also not affected by the mutations in GP5 [49].

### Virus particles with under-palmitoylated GP5 have no significant defect in cell entry

GP5 and M are the driving force for virus budding, but this does not necessarily exclude an additional function during cell entry that might be compromised in viruses having GP5 proteins with deleted acylation sites. One might assume that the total number of released particles of these GP5 mutants is comparable to those of wild-type PRRSV, but the overwhelming majority of them are not infectious, which would explain their greatly reduced titers. We therefore determined the specific infectivity of virus particles, defined as the ratio of infectious to

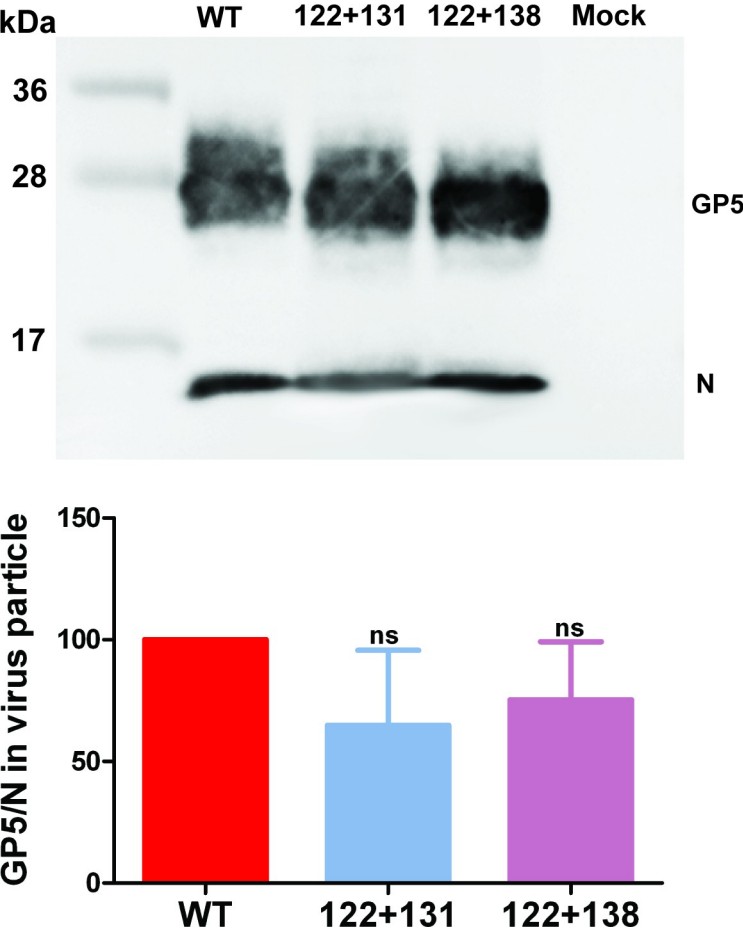

**Fig 8. Incorporation of wild type and under acylated GP5 into virus particles.** Amount of GP5 relative to N in virus particles. PRRSV WT or GP5 mutants with two exchanged acylation sites were grown for 48 hours in MARC-145 cell, pelleted from the supernatant through a sucrose cushion and analyzed by Western Blot with a mixture of antibodies against GP5 and N. Since the viruses grow to different titers the amount of the virus preparation loaded on the gel was adjusted to give the same band intensity for the GP5 protein. The optical density of GP5 and N bands were determined, the GP5 N ratio was calculated and normalized to wt particles (= 100%). The bottom panel shows the quantification of the result (mean ± standard deviation) from this and two other virus preparation. One-way ANOVA followed by Tukey's multiple comparison test revealed no significant difference between virus preparations.

total number of particles. We used the cleared and benzonase-treated supernatant of infected MARC-145 cells to assess the number of infectious particles by $TCID_{50}$ as well as the number of total genome-containing particles by qRT-PCR. Wild-type PRRSV exhibits a low specific infectivity of around 0.08% (Fig 9A, mean of three cell culture supernatants), but the variance between individual virus preparations was quite large and ranged from 0.005% to 0.23%.

We also evaluated the number of infectious particles with an infectivity assay. We incubated 100.000 genome-containing particles from the three virus preparations with ~$1x10^5$ MARC-145 cells or with ~$4x10^6$ porcine alveolar macrophages (PAMs), and visualized successful cell entry of individual virions by immunostaining of newly synthesized viral N protein. Since one replication cycle of PRRSV takes approximately 12 hours [50], cells were fixed at 10 hours after infection to ensure that immunopositive cells were a result of infection with the input virus and not by secondary spread. It was immediately obvious that MARC cells are less susceptible to infection than PAMs since the number of immunopositive cells were around 10times higher

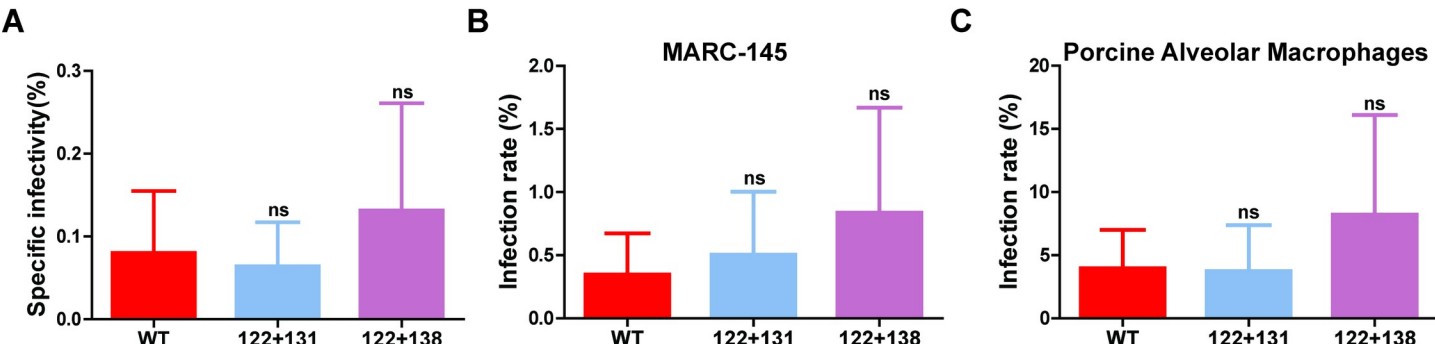

**Fig 9. Palmitoylation of GP5 is not required for cell entry of PRRSV.** (A) Specific infectivity of virus particles: The supernatant of MARC-145 cells infected at a MOI of 0.01 with PRRSV wildtype or mutants GP5 122+131 or GP5 122+138 were removed at 48 hours after infection. The cleared and benzonase-treated supernatant was used to determine the number of infectious particles with the TCID50 assay and the number of total virus particles by qRT-PCR. The specific infectivity (number of infectious particles divided by the number of total particles) was calculated for three different supernatants. (B +C) Cell entry assay: MARC-145 cells (B) or PAMs (C) were inoculated with 1x10⁵ particles as determined by qRT-PCR in paralell. Infected cells were visualized by immunofluorescence using antibodies against the PRRSV nucleocapsid protein (see S4 Fig) and counted. One-way ANOVA followed by Tukey's multiple comparison test revealed no significant difference between PRRSV wildtype and GP5 mutants.

in the latter if infected with the same virus preparation (S4 Fig). From the number of infected cells of three virus supernatants, we calculated a specific infectivity of 0,35% for MARC cells and 3,95% for PAMs (Fig 9B and 9C).

We then performed the same assays with the two acylation-deficient GP5 virus mutants. We calculated a specific infectivity of 0.064% using $TCID_{50}$ and of 0.5% and 3.74% with the cell entry assay using MARC cells and PAMs, respectively for GP5 122+131 and 0.13% (TCID 50) and 0.838% and 8.22% (cell entry assay for MARC cells and PAMs) for GP5 122+138 (Fig 9). However, the calculated numbers are not significantly different from wild-type PRRSV. We concluded that the infectivity of the mutants GP5 122+131 and GP5 122+138 are not reduced relative to wild-type PRRSV. Thus, the drastically reduced virus titers are not caused by a defect in virus entry, but rather by the total number of particles released from the cells.

## Discussion

### GP5 and M of PRRSV are palmitoylated at membrane-near cysteines

We report here that GP5 and M of PRRSV-1 and -2 strains are palmitoylated at highly conserved membrane-near cysteines, both if the proteins are expressed individually and in virus-infected cells (Figs 1 and 2). To block S-acylation completely, every cysteine in GP5 and M has to be exchanged indicating that each residue contains acyl chains (Fig 3A and 3D). However, removal of individual cysteines in GP5 had differing effects on overall palmitoylation of the protein. Removal of the first and last cysteine caused a reduction in palmitoylation of GP5 of both virus species, but results were not significantly different from wild-type GP5. In contrast, exchanging the cysteine in the centre of the triplet reduced GP5 acylation disproportionately, to around 30% in both viruses, which showed that lack of fatty acid attachment at this site also decreases acylation at another site (Fig 3E–3H). If the first and last cysteines were replaced simultaneously, GP5´s acylation levels were reduced to ~30% indicating that the central cysteine is still completely acylated and hence not affected by the two other sites. In contrast, if the central and the C-terminal cysteines were both exchanged, acylation was (almost) undetectable indicating that lack of acylation of the central cysteine prevented acylation of the first. Thus, there is cooperativity between the three acylation sites, the central cysteine being most important for complete acylation of GP5, especially at the first cysteine. In contrast, a fatty acid

attached to the C-terminal cysteine does not positively affect acylation of the other two sites and its acylation is only slightly increased by fatty acid attachment to the central residue. Such a synergistic effect between acylation sites was also observed for the M protein, since removal of one site decreased overall palmitoylation disproportionately, removal of Cys 99 or Cys 102 led to a decrease in palmitoylation to 20% and 35%, respectively (Fig 5D).

Cooperativity between two neighbouring acylation sites has also been reported for the ER-localized chaperone calnexin. Double acylated calnexin is resistant to cytosolic thioesterases, probably by triggering membrane imbedding of the fatty acids, which are rapidly cleaved from monoacylated calnexin. In concert with other protein modifications, cycles of acylation and deacylation regulate the stability and function of calnexin [51]. No turn-over of fatty acids has so far been detected in viral membrane proteins, including the HA of Influenza virus [43], and HA and other type 1 membrane glycoproteins are stoichiometrically acylated in virus particles [52, 53]. Thus, it seems unlikely that acylation of GP5 and M is dynamic, probably because the structural viral proteins perform their function only once, i.e. there is no need for regulation.

Since both M and GP5 are acylated if expressed alone (Fig 2), we conclude that monomers of GP5 and M are the substrate of acylation. Under those circumstances, the majority of Gp5 and M molecules remain in the ER and only a small subfraction has access to the cis-Golgi, which might be due to overexpression (S1 Fig), suggesting that acylation occurs in the ER. This is in contrast to palmitoylation of HA of Influenza A virus, where freshly acylated HA is already trimerized, but contains Endo-H sensitive carbohydrates indicating that the modification occurs in the early Golgi [43].

S-acylation of cellular proteins is mediated by one or several of the 23 members of the ZDHHC family of acyltransferases, polytopic membrane proteins that contain a DHHC (Asp–His–His–Cys) motif embedded within a 50-amino acid cysteine-rich domain (CRD) in one of their cytoplasmic domains. We have recently shown that HA of Influenza A virus, but not the hemagglutinating proteins of Influenza B and C virus, is acylated by DHHCs 2, 8, 15 and 20 [42]. They are exclusively localized in the Golgi (except DHHC 20, which is also in the ER), the intracellular site where acylation of HA occurs [43, 54]. However, GP5 and M are most likely modified by enzymes residing in the ER, and thus likely candidates are DHHCs 1, 4 and 6 [54], the latter two contain canonical KKXX-like motifs at their C-termini for ER retention [55, 56]. However, localization of DHHCs might differ between cell types and it is also conceivable that DHHCs residing only temporarily in the ER during their transit to other organelles of the exocytic pathway are involved in acylation of GP5 and M.

## Palmitoylation of both M and GP5 is required for virus growth

Using reverse genetics, we showed that acylation is required for virus growth, since no infectious virus could be rescued if either the three cysteines in GP5 or the two in M were replaced in the genome of XH-GD, a PRRSV-2 strain (Figs S2 and S4). Likewise, simultaneous exchange of Cys 131 and Cys 138, which completely prevented acylation of GP5 of the closely related VR-2332 strain, also inhibited generation of infectious virus. Viruses lacking individual acylation sites in GP5 could be rescued, but grew to significantly lower titers when one site was exchanged (~10- to 100-fold reduction) and was even more pronounced (1,000- to 100,000-fold) when two sites were replaced. Likewise, exchange of individual cysteines in M reduced virus titers by 10- (M99) or 1,000-fold (M102) (Fig 5).

The essential role of acylation is also reinforced by the notion that all GP5 and M sequences of PRRSV-1 and PRRSV-2 strains in the NCBI database contain the acylated cysteines (Fig 1A). Likewise, alignment of GP5 and M amino acid sequences of viruses from the *Arteriviridae*, such as equine arteritis virus (EAV), lactate dehydrogenase elevating virus (LDV) and

simian haemorrhagic fever virus (SHFV) revealed that all of them, except M of EAV, have cysteines located in the membrane-proximal part of the cytoplasmic tail (Fig 1B). Absolutely conserved is the central cysteine in the cytoplasmic tail of GP5, which has the strongest effect on overall palmitoylation of the protein. If no cysteine is present at the position identified as acylation site in GP5 and M of PRRSV, it is often replaced by a hydrophobic amino acid having a long and bulky side chain, such as Ile, Val and Phe which might functionally substitute a fatty acid. For the hemagglutinin of Influenza A virus, it was shown that acylated cysteines can be replaced with similar amino acids with only little effect on its functionality, whereas replacement by a serine prevented rescue of infectious virus [57]. Based on the sequence alignment one might speculate that an ancestral Arterivirus acquired the central cysteine in the cytoplasmic tail of GP5 that allowed it to hijack the cellular palmitoylation machinery which in turn increased virus replication. After separation into several genera individual viruses acquired additional cysteines at different positions in GP5 and also in the M protein that became palmitoylated and increased virus replication even further.

## Palmitoylation of GP5 affects virus budding

In order to locate which step of the viral replication cycle is affected, virus budding or virus entry, we investigated both steps in more detail. Removal of palmitoylation sites from viral glycoproteins often affect their fusion activity [58], but it is unknown for arteriviruses which glycoprotein complex harbours the viral fusion machinery. However, if removal of palmitoylation sites from GP5 would affect cell entry but not budding of viruses, the ratio of infectious to total particles in a virus population would be reduced. We, therefore, established two assays to estimate the specific infectivity of PRRSV particles: First we calculated the ratio of infectious particles as determined by $TCID_{50}$ to the total number of genome-containing particles as determined by qRT-PCR. The value we obtained is low (0.1%), meaning that roughly only one out of 1,000 particles is infectious. Then, we used 100,000 particles to infect MARC cells and PAMs and determined successful cell entry by immunofluorescence with antibodies against the viral nucleocapsid protein (S4 Fig). For MARC cells an infection rate of 0.3% was observed, but in macrophages it was 10 times higher, around 4% (Fig 9).

It is a long-standing observation that only a small fraction of (at least enveloped) virus particles produced by a cell are infectious. The reasons for this phenomenon include the failure to package the genome or a protein into particles, lethal mutations acquired in viral genes due to the high error rate of the viral polymerase and/or the lack of fully susceptible cell culture systems [59]. The latter explains the large difference in the infection rate between MARC cells and PAMs. Macrophages, but not MARC cells contain lectins at the cell surface and for sialoadhesin (also know as CD 169 or sialic acid-binding immunoglobulin-type lectin (siglec) 1) it has been shown that it binds to sialic acid residues attached to GP5 [14, 60]. Although knock-out of CD 169 expression does not affect infection of pigs with PRRSV and the pathogenesis of acute disease [61], other siglecs, especially siglec 10 might also mediate attachment of PRRSV [62].

When the same assays were done with mutant PRRSV particles, no statistically significant differences to wild-type PRRSV were observed (Fig 9). This is plausible, since the only known and at least in some cells non-essential function of GP5/M during virus entry is binding to heparan sulfate and to lectins at the cell surface. Since the ectodomains of both proteins carry out these activities, it seems unlikely that fatty acids attached to the cytoplasmic tails affect these processes [13, 14, 61].

The molecular mechanism of arterivirus budding is largely unexplored, but the essential role of GP5/M can be divided into several consecutive steps. GP5 and M must form a

disulfide-linked dimer as an essential prerequisite for their transport out of the ER to the budding site in the Golgi, where they are retained by so far unidentified signals. ER-localized monomers of GP5 and M are the substrate for acylation, but removal of palmitoylation sites from both GP5 and M had no effect on dimerization and subsequent trafficking to the medial Golgi, measured as acquisition of Endo-H resistant carbohydrates (Figs 6 and S3). Furthermore, non-acylated GP5 and non-acylated M co-localize in transfected cells in a perinuclear region to the same extent as acylated GP5 and M (Fig 7) further confirming the conclusion that acylation is not essential for intracellular trafficking of the GP5/M dimers.

Having shown that palmitoylation of GP5 and M does not affect dimer formation and transport to the budding site, we used qRT-PCR to investigate release of genome-containing particles into the supernatant of cells transfected with full-length cDNA clones. Cells transfected with cDNA encoding wild-type PRRSV and the GP5 mutant 122+131, which is able to generate infectious virus particles, release ~1000 and ~800 particles, respectively, into the cellular supernatant. In contrast, no genome-containing particles were detected for any of the mutant cDNAs, where the two acylation sites in M, three in GP5 or both together were removed, although the amount of intracellular viral genome copy numbers and GP3 messenger RNAs were the same in every transfected cell culture plate (Fig 5)

Next, it needed to be tested if GP5/M dimers are incorporated into the virion. We determined the amount of GP5 relative to the nucleocapsid protein N, but found no significant difference between virus particles lacking two acylation sites in GP5 and wild-type PRRSV (Fig 8). We concluded that GP5 does not need to be fully palmitoylated to be incorporated into virus particles. After budding, particles are transported through the Golgi and finally secreted by the cells. We think that a role of palmitoylation for virus egress is very unlikely, since the fatty acids are now located inside the virus particle and hence cannot interact with the cellular machinery that controls exocytosis.

In summary, we did not find evidence that M and GP5 lacking palmitoylation sites exhibit defects in cell entry or trafficking to the budding site. However, these mutants are unable to generate virus particles and hence we conclude that palmitoylation of GP5 and M is required for virus assembly and/or budding. This entails oligomerization of several GP5/M dimers and the induction of membrane curvature to form a bud and finally a spherical particle. There are several, not mutually exclusive possibilities how acylation might affect these processes. Acylation might induce partitioning of GP5 and M into cholesterol-rich nanodomains of the membrane that serve as viral assembly sites, similar to what has been proposed for the role of acylation of HA of Influenza virus, S of coronaviruses and other viral type 1 transmembrane proteins [34, 63]. The cholesterol content in the ER is low (5%), but successively increases throughout the Golgi until it reaches 40% in the trans-Golgi network [64]. One might speculate that acylated GP5/M dimers partition into Golgi-localized nanodomains upon arrival from the ER, which concentrate them to facilitate subsequent virus assembly. The fatty acids might then trigger oligomerization of GP5/M dimers possibly induced by interactions between the membrane-embedded domains of both proteins. The substantial number of hydrophobic moieties attached to GP5 and M is very likely to profoundly alter the proteins´ biophysical properties, and hence their ability to interact with membranes and with each other. Large-scale assembly of GP5/M dimers would create a coat within the lipid bilayer that finally helps to bend the membrane. Furthermore, protein-bound fatty acids might directly induce membrane curvature as demonstrated for HA of Influenza virus [65]. Any of these processes might be less effective if either GP5 or M have a reduced number of protein-bound fatty acids explaining the reduced number of particles released by infected cells.

Protein palmitoylation is also involved in virus budding of the closely related coronaviruses, which also occurs at membranes of the exocytic pathway. The minimal elements for the

formation of virus-like particles are the small and hydrophobic E-protein and the M-protein [66]. However, the coronavirus M protein, which is structurally (short ectodomain, three TMRs, long cytoplasmic domain) equivalent to M and GP5 of arteriviruses, is apparently not palmitoylated since no cysteine cluster is present in the membrane-near part of the cytoplasmic domain. Instead, besides the S protein, the E protein contains a palmitoylated cluster of cysteines in a membrane-proximal part of their cytoplasmic tails which is required for production of virus-like particles [33, 67]. Although the palmitoylated proteins of corona- and arteriviruses are structurally different, the mechanism how the fatty acids affect virus budding might be similar. Due to the hydrophobic effect, non-polar moieties attached to a membrane-proximal region of the proteins might exclude water molecules from the interface between the hydrophobic part of the lipid bilayer and the cytosol which in turn might facilitate spontaneous oligomerisation of palmitoylated hydrophobic proteins in the membrane. The resulting microassemblies might act as a nucleus for the recruitment of other viral components, which finally leads to virus assembly and budding. High-resolution fluorescence microscopy might be a useful tool to determine whether GP5 M assemblies exist and whether these clusters are reduced if palmitoylation sites are removed.

## Material and methods

### Ethics statement

The animal work was approved by the governmental agency, the Landesamt für Gesundheit und Soziales (LAGeSo) in Berlin, Germany (approval number T0297/17).

### Cells and viruses

Cell lines BHK-21 (baby hamster kidney cells, ATCC C13), 293T (Human embryonic kidney cells, ATCC CRL-3216), and MARC-145 (simian kidney epithelial cells derived from MA-104, ATCC CRL-6489) were maintained as adherent culture in Dulbecco's Modified Eagle's Medium (DMEM) (PAN, Aidenbach, Germany) supplemented with 10% fetal calf serum (FCS), 100U of penicillin per ml, and 100 mg of streptomycin per ml at 37˚C in an atmosphere with 5% CO2 and 95% humidity. Porcine alveolar macrophages (PAMs) were isolated from 10- to 11-week-old pigs by a lung lavage as described previously [68]. PAMs were maintained in RPMI medium (PAN, Aidenbach, Germany) containing 10% fetal calf serum (FCS) (Perbio, Bonn, Germany), 100U of penicillin per ml, and 100 mg of streptomycin per ml at 37˚C in an atmosphere with 5% $CO_2$ and 95% humidity.

PRRSV virus strains: Lelystad virus: low pathogenic PRRSV-1 prototype strain [3], Gen-Bank accession number: M96262.2; VR-2332: low pathogenic PRRSV-2 prototype strain [4], accession number: AY150564.1; XH-GD: Chinese highly pathogenic PRRSV-2 strain, accession number EU624117.1. The latter was rescued from the infectious cDNA clone pOKXH-GD as described below.

### Plasmids, mutagenesis and transfection of cells

Synthesis of GP5-HA of both VR-2332 and Lelystad virus and their cloning into the plasmid pCMV-TNT (Promega, (Mannheim, Germany), containing T7 and CMV promotors) has been described [47, 69]. The HA-tag consists of the amino acids YPYDVPDYA which is fused via a small linker (PV) to the C-termini of GP5 of both strains. M of both VR-2332 and Lelystad where fused at their C-termini by a small linker (GSG) to a 7xHis-tag and cloned into pCMV-TNT. Plasmid were used for site-specific mutagenesis using overlap extension polymerase chain reaction (PCR) as described previously [70]. Cells in 6-well plates were

transfected with 2.5 µg plasmid DNA using Lipofectamine 3000 (Thermo Fisher Scientific, Carlsbad, United States) as described by the manufacturer.

To analyze glycosylation of GP5, cells were lysed in glycoprotein denaturing buffer (0.5% SDS, 40 mM DTT), boiled for 10 min at 100˚C and an aliquot was digested either with endo-beta-N-acetyl-glucosaminidase (Endo H, 2.5-5units/µL, 4 h at 37˚C) or Peptide-N-Glycosidase (PNGase F, 2.5-5units/µL, 4 h at 37˚C) according to the manufacturer's instruction (New England Biolabs, Frankfurt am Main, Germany). Deglycosylated or untreated samples were supplemented with reducing SDS-PAGE loading buffer and subjected to SDS-PAGE and Western blot.

To analyze dimerization of GP5, virus-infected MARC-145 cells were lysed in 0,5% SDS, and quickly boiled in non-reducing (without mercaptoethanol) or reducing (with mercaptoethanol) SDS-PAGE loading buffer and subjected to SDS-PAGE and Western blot.

## Acyl-resin assisted capture (Acyl-RAC)

Protein S-acylation was analyzed by the Acyl-RAC assay as described [71] with some modifications. Transfected or infected cells in a six well plate were washed with PBS, lysed in 500µl buffer A (0.5% Triton-X100, 25 mM HEPES (pH 7.4), 25 mM NaCl, 1 mM EDTA, and protease inhibitor cocktail) per well. Disulfide bonds were reduced by adding Tris (2-carboxyethyl) phosphin (TCEP, Carl Roth, HN95.2) to a final concentration of 5 mM and incubated at room temperature for 45 min. Free SH-groups were blocked by adding methyl methanethiosulfonate (MMTS, Sigma, 64306, dissolved in 100 mM HEPES, 1 mM EDTA, 87.5 mM SDS) to a final concentration of 1.5% (v/v) and incubated for 4 h at 40˚C. Subsequently, three volumes of ice-cold 100% acetone was added to the cell lysate and incubated at -20˚C overnight. Precipitated proteins were pelleted at 5,000xg for 5 minutes at 4˚C. Pelleted proteins were washed four times with 70% (v/v) acetone, air-dried, and then re-suspended in binding buffer (100 mM HEPES, 1 mM EDTA, 35 mM SDS and protease inhibitor cocktail). 20–50 µl of the sample was removed to check for total protein expression by western blotting. The remaining lysate was divided into two equal aliquots. One aliquot was treated with hydroxylamine (0.5 M final concentration, added from a 2M hydroxylamine stock adjusted to pH 7.4) to cleave thioester bonds. The second aliquot was treated with 0.5M Tris-HCl pH 7.4. 30µl thiopropyl Sepharose beads (Sigma, T8387), which were beforehand activated by incubation for 30 min in aqua dest, were added at the same time to capture free SH-groups. Samples were incubated with beads overnight at room temperature on a rotating wheel. The beads were then washed 5x in binding buffer and bound proteins were eluted from the beads with 2x reducing SDS-PAGE sample buffer for 10 minutes at 95˚C. Samples were then subjected to SDS-PAGE and western blotting.

## SDS-PAGE and Western blot

After sodium dodecyl sulfate-polyacrylamide gel electrophoresis (SDS-PAGE) using 12% polyacrylamide, gels were blotted onto polyvinylidene difluoride (PVDF) membranes (GE Healthcare, Freiburg im Breisgau, Germany). After blocking of membranes (blocking solution: 5% skim milk powder in PBS with 0.1% Tween-20 (PBST)) for 1 h at room temperature, the following antibodies were applied overnight at 4˚C: rabbit-anti-HA tag antibody (ab9110, Abcam, Cambridge, UK, diluted 1:10.000 in blocking solution), mouse-anti-His tag antibody (Invitrogen, USA, 1:2000) were used to detect GP5-HA or M-His in plasmid transfected cells, monoclonal mouse-anti-GP5, M and N antibodies (reactive with PRRSV-2 strains, 1: 1000) were used to detect GP5, M and N protein in virus infected cells or virus particles. After washing (3x10 min with PBST), the following horseradish peroxidase-coupled secondary antibody were applied for 1 hour at room temperature: anti-rabbit (Sigma-Aldrich, Germany, 1:5.000)

or anti-mouse (Bio-Rad Laboratories, USA, 1:2000). After washing, signals were detected by chemiluminescence using the ECLplus reagent (Pierce/Thermo, Bonn, Germany) and a Fusion SL camera system (Peqlab, Erlangen, Germany).

## Immunofluorescence and confocal microscopy

Transfected BHK-21 or infected MARC-145 cells, grown in 6-well or 24-well plates (Sarstedt, Nümbrecht, Germany), were washed twice with PBS, fixed with paraformaldehyde (4% in phosphate-buffered saline (PBS)) for 20 min at room temperature, washed twice with PBS, permeabilized with 0.2% Triton in PBS for 10 min at room temperature, and washed again twice with PBS. After blocking (blocking solution: 3% bovine serum albumin (BSA) in PBS with 0.1% Tween-20 (PBST)) for 30 min at room temperature, the cells were incubated with the following antibodies for one hour at room temperature: rabbit-anti-HA tag antibody (ab9110, Abcam, Cambridge, UK, 1:500), mouse monoclonal anti-HA tag (6E2) antibody (2367S, Cell signaling, USA, 1:100) or monoclonal mouse-anti-M antibodies (reactive with PRRSV-2 strains, 1:100) were used to detect GP5-HA or M-His in plasmid transfected cells, monoclonal mouse-anti-GP5 and M antibodies (reactive with PRRSV-2 strains, 1:100) or mouse anti-N antibody (reactive against PRRSV-2, 1:1000, Creative Diagnostics, USA) is used to detect GP5, M and N in infected MARC-145 cell, rabbit anti-GM130 antibody (ab52649, Abcam, Cambridge, UK, 1:200) is used for staining the cis-Golgi. After washing three times with PBS, cells were incubated with secondary antibody (anti-mouse or anti-rabbit IgG (H+L) from goat coupled to Alexa Fluor 488 or 568, Invitrogen, Darmstadt, Germany, 1:1000). Pictures were recorded using a ZEISS Axio Vert. A1 inverse epifluorescence microscope or the Nikon inverted microscope eclipse Ti (Visitron Systems GmbH) for confocal pictures.

## Reverse genetics and mutagenesis of full-length PRRSV cDNA clone

The full-length infectious PRRSV cDNA clone pOKXH-GD (a kind gift of Prof. Guihong Zhang, South China Agricultural University) was constructed from the XH-GD PRRSV strain (GenBank accession no. EU624117) and vector pOKq [72]. The entire genome of PRRSV is present in two plasmids: the plasmid pOK-A contains a CMV promotor plus the PRRSV nucleotide sequences 1 to 6465 (= fragment A) and pOK-B the nucleotide sequences 6466 to 15345 (encoding most of the structural proteins) plus a terminal BGH RNA transcription terminator sequence (= fragment B). Site-directed mutagenesis on the pOK-B plasmid using overlap extension polymerase chain reaction (PCR) was described previously [70]. To avoid the possibility of reversion back to the wild type sequence two nucleotides were exchanged to introduce the same amino acid exchanges as in GP5-HA and M-His of VR-2332. The GP5 and M genes in the pOK-B plasmids containing the desired mutations were sequenced to exclude unspecific second site mutations introduced by PCR. To reconstruct the full PRRSV genome fragment A was cut from the pOK-A plasmid by digestion with AfI and NotI and ligated with the pOK-B plasmid digested with the same enzymes. The complete plasmid was also constructed from wild type pOK-B in parallel. After ligation the resulting plasmids were digested with two different restriction enzymes. The resulting band pattern was identical between mutants and wild-type indicating that the procedure did not introduce deletions or insertions into the PRRSV cDNA.

The plasmids (2.5 µg) containing the complete PRRSV genome were transfected into 80% confluent BHK-21 cells grown in 6-well plates using Lipofectamine 3000 (Thermo Fisher Scientific, Carlsbad, United States) as described by the manufacturer. After 48 hours (in two other attempts at 72 or 96 hours after transfection) the supernatant was removed, and cells were subjected to immunofluorescence assay using anti-GP5 monoclonal antibody. The

supernatant was cleared by low speed centrifugation and 500µl was used to infect (PBS washed) MARC-145 cells grown to 70–80% confluence on 6-well plates. After incubation for 1.5 h at 37˚C, the inoculums were removed, cells were washed twice with PBS, and cells were incubated in culture medium (DMEM with 2% FCS) for 48 h. Cells were then subjected to immunofluorescence assay using anti-GP5 monoclonal antibody.

The authenticity of rescued viruses was confirmed by Sanger-sequencing of a PCR-product encompassing the mutation site, and viruses were further passaged 2–3 times in T175 flask to generate a for virus stock for subsequent experiments. The presence of the mutation was again verified by Sanger-sequencing.

## Virus titration

Sub-confluent MARC-145 cells in 24-well plates were infected with rescued viruses (P5) at a multiplicity of infection (MOI) of 0.001. After 1.5 h incubation at 37˚C, cells were washed three times with PBS and incubated at 37˚C in 0.5 ml DMEM containing 2% FCS in a $CO_2$ incubator. At certain time points (12, 24, 48, 72 and 96h) post-infection, supernatants were collected and frozen at -80˚C until use. The viral titers were determined in MARC-145 cells with the tissue culture infection dose 50% endpoint assay (TCID 50).

## Preparation of PRRSV particles

MARC-145 cells grown in eight T-175 flasks were infected with either PRRSV XH-GD wild-type or rescued cysteine-mutants at a multiplicity of infection (MOI) of 0.001 and incubated for 3–4 days at 37˚C in DMEM until cytopathic effects became visible. Cell culture superna-tants were harvested, cleared by low-speed centrifugation (3,000g, 30 min) and applied on top of a sucrose cushion (20% w/v in TNE buffer (10 mM Tris-HCl, 10 mM NaCl, 1 mM EDTA, pH 7.5)). Samples were ultracentrifuged for 3 h at 28,000 rpm at 4˚C in a Beckman SW32 rotor. The virus pellet was resuspended in TNE buffer and frozen at -80˚C until use.

## RNA isolation and quantification of genome copy numbers by quantitative RT-PCR

Quantitative RT-PCR was performed to calculate the amount of viral genomic RNA (as a mea-sure of total number of virus particles) released into the supernatant of transfected BHK cells ($\sim 1 \times 10^6$ cells) or infected MARC cells grown in 6-well cell culture plates. Cell supernatants (1ml) were cleared by low speed centrifugation (30min, 5000g) and a 400µl aliquot was treated with 75U/ml Novagen benzonase nuclease (Merck Millipore, Darmstadt, Germany) for 20 hours at 37˚C to remove remaining plasmid DNA from transfected BHK cell supernatants and for 30 minutes to remove free (not virus-associated) genomic RNA from infected MARC cell supernatants. To extract RNA from supernatants the innuPREP virus TS RNA Kit (Analytik Jena, Germany) was used. To extract RNA from transfected BHK cells the RNeasy Plus Mini Kit (QIAGEN, Germany) was used. 10ul RNA was then reverse transcribed into cDNA using High Capacity cDNA reverse transcription kit (Applied Biosystems, Carlsbad, United States). From the resulting 20 µl cDNA sample 3 µl was used for qRT-PCR, which was performed with the StepOnePlus real-time PCR system (Applied Biosystems) with SYBR green as fluorophore. For quantification of viral genome copy numbers, a standard curve was generated by serial dilution ($10^8$–$10^1$ copies/µl) of the full-length wt cDNA plasmid. The primers amplified a 180 bp fragment (427bp - 606bp) from the GP3 gene of PRRSV, which is also present in the mes-senger RNA. Gene copy numbers were calculated with StepOne Software v2.3 (Applied Biosystems).

## Virus entry assay

MARC-145 and porcine alveolar macrophages (PAM) cells were infected with $10^5$ virus particles (as determined by qRT-PCR) of wild-type PRRSV or mutants GP5 122+131 or GP5 122 +138. Since one replication cycle of PRRSV takes 12 hours [50], cells were fixed 10 hours post-infection (p.i.) with 4% paraformaldehyde (PFA) at 4°C and permeabilized with 0.2% Triton X-100 prior to immunofluorescence staining with primary monoclonal antibodies against PRRSV nucleocapsid protein (reactive against PRRSV-2, 1:1000, Creative Diagnostics, USA). As secondary antibody Alexa Fluor 488 goat anti-mouse IgG(H+L) (Invitrogen, Darmstadt, Germany, 1:1000) was used. Cell nuclei were stained with DAPI. The infected cells in a 6-well platte were counted, either completely (MARC cells) or five random fields which corresponds to $1x10^5$ cells and the total number was calculated from that number (PAMs). Pictures were taken with a ZEISS Axio Vert. A1 inverse epifluorescence microscope.

## Statistical analysis

Data were expressed as means ± standard deviations. The significance of the variability among the trials was determined by one-way or two-way ANOVA analysis of variance as described in the figure legend using GraphPad Prism (version 5.0) software.

## Supporting information

**S1 Fig. Colocalization of GP5 and M with cis-Golgi marker.** BHK-21 cells were transfected with GP5-HA wt (upper panel) or M-His wt (lower panel). GP5 was stained with monoclonal antibodies against the HA-tag, M with monoclonal anti-M antibodies in both cases followed by secondary anti-mice antibodies coupled to Alexa Fluor-488. The cis-Golgi was stained with polyclonal antibodies against GM130 followed by secondary anti-rabbit antibodies coupled to Alexa Fluor-568 and the nuclei by DAPI. Right panel: Co-localization of M or GP5 with GM 130 from at least 40 cells was quantified with the Pearson's correlation coefficient method using the JACoP plugin of the ImageJ software.
(TIF)

**S2 Fig. Acylation of both GP5 and M is essential for virus replication.** (A) Analysis of mutants where all cysteines in GP5 and M are exchanged. BHK cells were transfected with the viral genome of the XH-GD strain (wt) or with the genomes of mutants where the three cysteine in GP5 (pGP5 SSS) or the two cysteines in M (pM SS) or cysteines in both proteins (pGP5 SSS+M SS) were exchanged to serine. After 48 hours cell supernatants were removed and used to infect MARC-145 cells, which were processed for immunofluorescence 48 hours later. Transfected and infected cells were permeabilized and stained with anti-GP5 monoclonal antibody and Alexa-568 anti-mouse secondary antibody and the nuclei with DAPI. (B) Analysis of mutants where one or two cysteines in GP5 and one in M were exchanged. One cysteine in M (pM 99, pM 102, one cysteine in GP5 (pGP5 122, GP5 131, GP5 138) or two cysteines in GP5 (GP5 122+131, GP5 122+138, GP5 131+138) were exchanged to serine. MARC-145 cells were infected with the supernatant from transfected BHK-21 cells and 48 h later stained with anti-GP5 monoclonal antibody and Alexa-568 anti-mouse secondary antibody and the nuclei with DAPI. Note that only the double mutant GP5 C131+138, that is not palmitoylated in the closely related VR-2332 strain (Fig 2H) could not be generated.
(TIF)

**S3 Fig. Acquisition of Endo-H resistant carbohydrates of wt and non-acylated GP5 when expressed alone or together with M.** BHK-21 cells were transfected with plasmids encoding GP5-HA wt, non-acylated GP5-HA SSS, M-His wt or non-acylated M-His SS from the VR

2332 strain. 20 hours after transfection cells were lysed and digested with Endo-H or left untreated as indicated. Samples were subjected to SDS-PAGE and western-blotting with anti-HA or anti-His antibodies.
(TIF)

**S4 Fig. Infection of MARC cells and PAMs with wild-type PRRSV and acylation deficient GP5 mutants.** $1x10^5$ Marc cells or $4x10^6$ PAMs were infected with $1x10^5$ virus particles as determined by qRT-PCR. 10 hours after infection cells were fixed, and stained with mouse anti-N antibody followed by anti-mouse IgG antibody coupled to Alexa 568. Pictures were recorded using a ZEISS Axio Vert. A1 inverse epifluorescence microscope.
(TIF)

## Author Contributions

**Conceptualization:** Michael Veit.

**Formal analysis:** Minze Zhang.

**Funding acquisition:** Klaus Osterrieder, Michael Veit.

**Investigation:** Minze Zhang, Xiaoliang Han.

**Writing – original draft:** Michael Veit.

**Writing – review & editing:** Minze Zhang, Klaus Osterrieder.

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
