## [Decision Letter · Decision Letter 0]

22 Dec 2020

Dear Dr. Veit,

Thank you very much for submitting your manuscript "Palmitoylation of the envelope membrane proteins GP5 and M of porcine reproductive and respiratory syndrome virus is essential for virus growth" for consideration at PLOS Pathogens. As with all papers reviewed by the journal, your manuscript was reviewed by members of the editorial board and by several independent reviewers. In light of the reviews (below this email), we would like to invite the resubmission of a significantly-revised version that takes into account the reviewers' comments. Please make sure to respond to all the reviewer's comments.

We cannot make any decision about publication until we have seen the revised manuscript and your response to the reviewers' comments. Your revised manuscript is also likely to be sent to reviewers for further evaluation.

Sincerely,

Jens H. Kuhn

Associate Editor

PLOS Pathogens

Carolina Lopez

Section Editor

PLOS Pathogens

Kasturi Haldar

Editor-in-Chief

PLOS Pathogens

orcid.org/0000-0001-5065-158X

Michael Malim

Editor-in-Chief

PLOS Pathogens

orcid.org/0000-0002-7699-2064

Reviewer's Responses to Questions

**Part I - Summary**

Reviewer #1: This manuscript by Zhang and colleagues explores the functional impact of palmitoylation at key conserved cysteine residues in the PRRSV membrane proteins GP5 and M. The authors show that, indeed, conserved cysteines in GP5 and M are palmitoylated, and that some cysteines were more efficiently palmitoylated than others. Furthermore, viruses lacking all palmitoylated cysteines in GP5 or M could not be rescued, while single/double cysteine mutant viruses tended to grow to lower titers, suggesting the importance of these conserved amino acids for virus particle production. While the data is interesting, and have the potential to improve our understanding PRRSV assembly, I feel that the study falls short of providing convincing mechanistic evidence for why palmitoylation mutant viruses are defective.

Reviewer #2: This submission presents a study of the five to six cysteines on PRRSV GP5 and M proteins that are potentially subject to post-translational palmitoylation. The investigators documented that the cysteines are indeed palmitoylated. They then found that C to S substitutions, and therefore, the complete absence of palmitoyl adducts on GP5 and M, was incompatible with PRRSV production. Under-palmitoylation did, however, allow for virus production and the authors exploited this finding. They found that wild-type palmitoylation, relative to under-palmitoylation, allowed for more virus particle output, most likely due to increased virus budding. Notably the under-palmitoylation had little if any effect on virus entry. The roles for palmitoylation in virus budding, therefore, were discussed relatively thoroughly in this report.

Overall, the report appears to be valuable. It is important to know how viral protein palmitates operate in virus infections, and this study points sensibly to palmitoylation requirements in virus budding or some other aspect of particle morphogenesis. This reviewer’s opinion is that the report is well written, the experiments executed well, and the solid results generally interpreted properly. There are, however, still some questions about interpretation of findings and evidence for the conclusions made in the report. These questions are put forward in the comments below.

Reviewer #3: In the paper ‘Palmitoylation of the envelope membrane proteins GP5 and M of porcine reproductive and respiratory syndrome virus (PRRSV) is essential for virus growth’, Zhang et al. demonstrated that the 3 cysteines of the cytoplasmic tail of GP5 and the 2 cysteines of the cytoplasmic tail of M are palmitoylated (upon transfection and infection). If all cysteines of GP5 or M were replaced no virus could be rescued, demonstrating that acylation is essential. Trafficking of GP5 and M to the Golgi apparatus was not disturbed by the absence of acylation. These results were obtained with a technology that was successfully used before by the senior scientist with other viruses. Therefore, the conclusion that the multiple fatty acids attached to GP5 and M are crucial in the virus assembly can be supported. Since this is a lacking piece of the PRRSV replication puzzle, it is quite interesting. I have more reservations with the technology that was used for demonstrating that the palmitoylation is not important for the entry of the virus. GP5/M is important for the infection of receptor (siglecs)-carrying macrophages. This is not the case for Marc cells. As a result, a virus titer of a wild type virus stock in macrophages is tenfold higher than in MARC-145 cells. This explains the big difference between the TCID50 and genome copies, as also demonstrated in the present study. In order to study the role of palmitoylation of GP5 and/or M it is important to use macrophages that express receptors that interact with GP5/M. Furthermore, it is important to note that RT-qPCR does not only detect genomes inside virions but also free genomes. It would have been better to first eliminate the free genomes by benzonase treatment.

**Part II – Major Issues: Key Experiments Required for Acceptance**

Reviewer #1: 1. Fig. 1 A: PRRSV-1 strain Lelystad and PRRSV-2 strains VR-2322 and XH-GD serve as prototype references strains. This feels like a very surface level analysis. Given that considerable genetic variation exists between different PRRSV isolates, the authors could strengthen this analysis with a broader panel of sequenced PRRSV isolates. Furthermore, are these conserved cysteines found in other arteriviruses (EAV, LDV, SHFV, etc.). Is this conservation unique to GP5/M of all arteriviruses? Are there exceptions? This information should be provided up front, rather than waiting until the discussion section.

2. RE: “Palmitoylation of M and GP5 was also detected during replication of both viruses in isolated porcine alveolar macrophages, the target cells of PRRSV in pigs (not shown).” PAMs are a more relevant model system than monkey kidney cells (MARC-145), yet that data in PAMs is “not shown.” Either this data should be presented, or this sentence should be removed.

3. “…in contrast to the diffuse bands of GP5 in virus infected cells.” In a later figure, GP5 shows up as a double-band under similar reducing conditions. Is this an exposure issue? Are these “diffuse bands” being overexposed in the western blot for 1B compared to 1C? The authors should carefully compare viral infections (which express both GP5/M) to transfections with both GP5/M genes, as well as each gene alone.

4. RE: “The results suggest that GP5 and M remain in the ER if expressed alone, similar to what has been reported for…” To make this statement, the authors should repeat the experiment, but with co-transfection of both GP5/M. I think the authors are making too great of a leap here so early in the paper, and this whole paragraph seems out of place. The difference in banding patterns could be due to differences between cell types used, for instance, as N-glycosylation is known to be cell type dependent. However, they then go on to do IF to test their assumptions, but for some reason they switch from MARC-145 (infection) to 293Ts (transfection), and then to making assumptions on GP5/M co-localization with cellular markers in a completely different tissue/species cell line (transfected BHKs) (Supplementary Fig. 1). There should be consistency between the cell types used in these experiments. Further, the authors are measuring co-localization with a golgi marker and making assumptions about ER localization? The authors could easily test this assumption by repeating the experiment, but with an ER marker.

5. Fig 2. I appreciate the addition of Flotillin 2 as an internal assay control. This should be “flotillin” not “flotilin" though. It is correct in the text, but incorrect in the figure. Where is the flotillin blot for 2G? Aesthetically speaking, the images should all be cropped similarly. Sometimes the ladder is shown, sometimes it isn’t. Sometimes the 36 kDa band on the gel is shown, sometimes it isn’t. Most blots have a mock, but panel F doesn’t. The consistency in the figure is lacking.

6. Fig. 2. When each cysteine is replaced, there is a qualitative difference the amount of GP5 pulled down. This is a really interesting finding, especially given that some cysteines seem to be more important than others in terms of their relative palmitoylation state. I’m really curious why the same wasn’t done for M? Similar experiments for GP5 and M are carried on throughout the paper, but for some reason, not here.

7. Fig 2 legend. I’m curious about how the statistics were performed. If each independent experiment was normalized to GP5 wt set at 100%, and then that data was combined, your wt would be 100% with no statistical deviation (as presented in the graph). Then, wouldn’t any statistical comparison you make between your mutants and wt here show a significant difference (given the lack of error associated with the wt values)? It seems like the combined, non-normalized data, from three experiments (ratio of +HA to TE; which would also include the variation in the GP5 wt between experiments) would be the dataset that the statistics should be tested on. Not the normalized data. The authors need to be more specific in how their stats were analyzed in the legend and correct if need be.

8. It would be nice to see more consistency throughout the paper. Figure 1 has Lelystad (PRRSV-1) and VR2322 and XH-GD (PRRSV-2). XH-GD is dropped in figure 2, but then viral infections were done with XH-GD through the rest of the paper. For consistency purposes, it would have been nice to also see data for XH-GD in Fig 2 so analogies could be drawn with the same mutants used in transfection/infection experiments.

9. Fig. 3 results section:

a. In the text, viral cDNA was transfected into 293T cells, but in the legend and figure, it states that BHKs were transfected. So, which cell line was it?

b. It is unclear why this transfection step was necessary. If MARC-145 cells “support virus replication,” why then couldn’t all of this been done in those cells? Could the authors briefly provide more experimental details here?

c. “…five further attempts to rescue infectious mutant virus also failed.” It is unclear what the parameters were here. Did you do the same thing, five different times, or did you troubleshoot it differently? This sentence alone does not provide much help to the reader.

10. Fig. 3: A few points…

a. It would be nice to see the nuclear staining in all images. Just showing a black box isn’t informative.

b. Is there a reason why GP5 staining, but not M staining, is shown? It would be important to show that mutagenesis did not disrupt the ability of these viruses to express both of these proteins.

c. “…we could not detect viral genomes for any of the GP5 and M mutants in the supernatants of infected cells, whereas ~107 genome copies of wild-type virus were present.” I feel that this is the most important point here for Fig 3. There’s no point to show IF infection of MARC-145 cells if no virus particles were produced in the first place. Thus, showing infection via immunofluorescence in MARC-145 cells seems meaningless. Further, for the qPCR data, for all viruses in 3B, are similar genome copy numbers detected in the cell supernatant? This would be an interesting addition to this figure.

11. Fig. 3A. For the GP5/M palmitoylation mutants, where no viral genomes were detected in the supernatant: The authors should explore the fate of virions in the producer cells. Do viral genomes still accumulate in the cell? Can you detect intracellular viral particles? Do these proteins get stuck in different organelles, unlike the wild-type proteins which would be expected to traffic normally? More mechanistic underpinnings here would heighten the impact of this work.

12. Fig. 4. I appreciate seeing the growth kinetics for the wild-type and mutant viruses. As an important control, are GP5 and M in the mutants being incorporated into virus particles to similar levels as the wild-type proteins?

13. Fig. 4. The authors need to address the variability between the replicate experiments. I’m very surprised (and nervous) to see that sometimes there are several logs difference noted between wt and mutant (let’s look at 4B wt vs purple line @ 24h), yet there is no statistical difference between those points. This is a massive amount of experimental variation. The authors cannot conclusively say “each mutant virus grows to lower titers when compared to those of wild-type PRRSV” when all mutant viruses (except 122+131 and 122+138) reach the same endpoint titer that is not statistically different than the wild-type virus. This seems to be a misrepresentation of the data, and that selling point is made multiple times throughout the paper. This is one of the most important figures in the paper, yet the data (statistically speaking) doesn’t fit with the authors interpretation.

14. Fig. 4 methods: “the authenticity of rescued viruses was confirmed by sequencing, and viruses were further passaged 2-3 times in T175 flask to generate a for virus stock for subsequent experiments.” Was this sequencing performed before, or after virus passaging to generate stocks? How was it performed? Sanger sequencing of a PCR product? Sanger of multiple independently cloned PCR inserts? Next-gen sequencing? Critical details are missing here. I’m curious how stable these mutations were in their virus stocks, and how stringent the authors were in sequencing their stocks is important for data interpretation.

15. Fig. 4D. I don’t see the point in just showing the palmitoylation data for M single mutants 99 and 102, in comparison to wild type. If you want to present this level of detail, why then isn’t this being presented for all of the mutant viruses in A and B? I don’t see any rationale for why these two mutants were chosen. Further, the authors state 10 to 1000-fold differences at 48 h, but in the figure, non-significant differences at 12, 24, 72, and 96 hours are noted. This seems like only select information is being highlighted, rather than telling the whole story. The endpoint titer of these mutant viruses are not significantly different than wt, despite the “tendency for lower titers.”

16. Fig. 5. How have the authors controlled for equal protein loading in these westerns? No loading controls are shown. For the ratios, I would have expected each band in -ME and +ME to have been internally normalized to a loading control (on the same blot), and then those band intensities could be compared -ME to +ME. Why haven’t the authors similarly probed for M protein in these blots, thereby confirming that the 38 kDa band was a GP5-M heterodimer, rather than speculating? Further, and again to touch on consistency, why is the ladder shown in some but cropped out in others?

17. Connecting Fig. 3 and Fig. 5. No infectious virus is produced in the pGP5 SSS, pM SS, or pGP5 SSS + M SS mutants. In Figure 5, the authors suggest that GP5 dimerizes with M (non-reducing gel, although M blotting is not shown). Is the reason for the lack of secreted pGP5 SSS, pM SS, or pGP5 SSS + M SS viruses because these mutant GP5/M proteins do not dimerize together, thus, budding is impeded? If you pull down GP5, can you detect M and vice versa? More mechanistic details here would add additional strength to this work.

18. I would have appreciated Fig. 5 more if the authors would have better described now N-glycan addition and Endo-H protection fit with their interpretation that the “same number of molecules have passed through the medial golgi-complex.” I’m not fully understanding how the data fit with their interpretations. Importantly, “…Endo-H resistant bands were also detected if non-acylated GP5-SSS and M-SSS were co-expressed (data not shown).” Why is this data not shown? The authors lack experimental detail for why these mutants failed to make infectious virus. Does this, along with the co-localization experiment (Fig. 6), suggest it isn’t due to any trafficking issues?

19. Fig. 6. This figure analyzes localization of GP5/S mutants, for those mutants that could not produce infectious virus (Fig. 3). How does co-localization of GP5 and M in the golgi allow the authors to conclude that they are indeed dimerized? This is also an overexpression scenario, would is known to cause unintended effects on protein localization.

20. Fig. 7. I appreciate that the authors showed incorporation of mutant GP5 proteins in relation to wildtype. How was the blotting normalized for these concentrated viral preparations, especially since the mutant viruses tend to produce lower viral titers than wild-type? Was it by protein input? Careful descriptions are necessary here. Also, why were these particular mutants chosen but not others? In my opinion, this is an important control figure that should be shown for all viruses produced, and it should have been presented several figures back.

21. RE: Virus particles with under-palmitoylated GP5 have no significant defect in cell entry. This whole section was very difficult to follow. Two different assays, with two different viruses are being placed into a single table. The table headers aren’t easy to interpret on their own. The infection values are extremely low, and if you are looking at entry, waiting 10 hours to detect N seems very long. I get the point that the authors are trying to make, and it’s an important one, it just doesn’t come off as easily interpretable in the way the data is described and presented. Also, no staining for N is shown (which is integral to interpreting this entry assay), making this experiment difficult to review. If the authors could get a hold of a reporter PRRSV isolate, it would greatly improve their experimental setup here. It could be normalized by genome content, TCID50 could be determined by flow cytometry, and then entry measured by reporter gene expression shortly after infection.

22. Overall, I don’t see experimental data to explain why palmitoylation mutant viruses behave differently than wild type. To strengthen this paper, I feel that the authors need to delve deeper into that mechanism. To summarize the major figure conclusions:

a. GP5/M are palmitoylated (Fig. 1)

b. Some cysteines are more important for palmitoylation than others (Fig. 2)

c. Removing all cysteines in GP5 or M prevents viral particle release (Fig. 3)

d. Palmitoylation mutants tend to have lower titers (Fig. 4)

e. There is no difference in glycosylation or Endo H protection (Fig. 5)

f. GP5/M mutants co-localize together (Fig. 6).

g. GP5 mutants incorporate similarly to wt into virions (Fig. 7).

h. No defects in entry are noted (Table 1).

None of this data provides mechanistic insights into why GP5/M need to be palmitoylated, other than reduced palmitoylation of GP5/M has a tendency to impact viral titers.

23. The authors speculate that acylation affects the budding process. Could the authors use an ELISA to prove this point? If less viruses are budding out, this should be quantifiable in a protein ELISA, which can then further be compared to viral genome copies in infected cell supernatant.

24. Figure 8 is not sufficient as a standalone figure, and I would suggest that these alignments be added to the first figure.

Reviewer #2: 1. The report suggests a role for palmitate modifications in virus budding and virus particle membrane curvature. These are reasonable suggestions and the role in budding is somewhat supported by excluding other roles for the palmitoylations. However, there is not much direct evidence for a role for the palmitates in budding. One suggestion is to perform EM of the GP5 2-palm-deficient viruses. Any evidence of misshapen particles, deformed because under-palmitoylation creates aberrant membrane curvatures?

2. It is hard to accept one of the main conclusions of the paper regarding palmitates in virus budding. “We concluded that palmitoylation of GP5 does not affect incorporation of the GP5/M dimer into virus particles.” This appears to be based on the single palmitoylated GP5 mutant, lacking two palm cys residues. Any analysis of un-palmitoylated GP5 in virus incorporation could not be performed because palm-negative viruses were not obtained. Therefore, how can it be concluded that palmitoylation of GP5 is not required for incorporation into virus particles?

3. The PRRSV cDNAs encoding palm deficient GP5 and M can produce viral proteins but do not release infectious particles. Can the cDNA – transfected cells be complemented with GP5 and M expression vectors, to then produce viruses? Results from such complementation tests would further confirm findings in this study.

4. Virus egress via exocytosis is required after budding. Have roles for palmitates on GP5 and M in forming egress-competent viruses been considered, or excluded from consideration for some reason?

Reviewer #3: 1° See under Summary

I have reservations with the technology that was used for demonstrating that the palmitoylation is not important for the entry of the virus. GP5/M is important for the infection of receptor (siglecs)-carrying macrophages. This is not the case for Marc cells. As a result, a virus titer of a wild type virus stock in macrophages is tenfold higher than in MARC-145 cells. This explains the big difference between the TCID50 and genome copies, as also demonstrated in the present study. In order to study the role of palmitoylation of GP5 and/or M it is important to use macrophages that express receptors that interact with GP5/M. Furthermore, it is important to note that RT-qPCR does not only detect genomes inside virions but also free genomes. It would have been better to first eliminate the free genomes by benzonase treatment.

2° Macrophages are the real target cell. Therefore, some findings in MARC-145 cells should be confirmed in macrophages such as the demonstration that palmitoylation is not required for dimerization of GP5 and M, for transport to the Golgi apparatus and for incorporation into virus particles.

**Part III – Minor Issues: Editorial and Data Presentation Modifications**

Reviewer #1: 1. The manuscript should be updated with line numbers to help the reviewer address specific lines in the paper.

2. Abstract: “no virus could be rescued if all three cysteines present in GP5 or both present in M were replaced in a PRRSV-2 strain, indicated that acylation is essential for virus replication.” I don’t think “replication” is the correct term here. No viral genomes were released for these mutants, but the authors did not show that these mutants had any defect in viral genome replication in the producer cells. Perhaps, viral “egress from the cell” would be more suitable?

3. Fig 1 legend: “Note that GP5 also contain cysteines in the middle of the TMR.” These cysteines are immediately adjacent to the cytoplasmic tail. Is “cysteines in the middle of the TMR” correct here?

4. “…we exchanged them with serine in GP5-HA and M-His in both the VR-2322 and Lelystad viruses.” And here: “…revealed a remaining palmitoylation level of ~20% in the Lelystad and of only ~10% in the VR-2332 virus.” The way these sentences are worded makes it sound like you made these serine mutations in the viruses themselves, rather than expression vectors. Please double-check and correct this wording throughout.

5. Fig. 3 legend and throughout: “Alex-568” should be Alexa-568.

6. The manuscript should be thoroughly re-read and corrected for grammar and spelling errors.

Reviewer #2: 1. A point about terminology here: The term “replication” often refers to replication of viral genetic material. In this context, then, it is not established that the absence of palmitates eliminates PRRSV “replication”. In fact, replication appears to be demonstrated in the absence of palmitoylated GP5 – M. Palmitates instead are suggested to be needed for particle assembly or budding or perhaps particle egress. The use of the terms “replication” and “replication deficient” might be reconsidered in this manuscript.

2. Regarding consensus palmitoylation motifs, are there not predictive algorithms for palmitoylation? (intro stating lack of consensus motifs suggests otherwise).

3. Fig. 5; were thiol blockers (NEM or IAA) included during the cell lysis process, to prevent artifactual post-lysis disulfide formations?

4. The review process would be more effective if the text included line numbers, so if there is any additional review, a submission with line numbers would help.

Reviewer #3: Author summary - last sentence: ‘They might trigger assembly of GP5/M dimers to from…’ change into ‘They might trigger assembly of GP5/M dimers to form…’

PLOS authors have the option to publish the peer review history of their article (what does this mean?). If published, this will include your full peer review and any attached files.

Reviewer #1: No

Reviewer #2: No

Reviewer #3: **Yes: **Hans Nauwynck
---

## [Decision Letter · Decision Letter 1]

15 Mar 2021

Dear Dr. Veit,

Thank you very much for submitting your manuscript "Palmitoylation of the envelope membrane proteins GP5 and M of porcine reproductive and respiratory syndrome virus is essential for virus growth" for consideration at PLOS Pathogens. As with all papers reviewed by the journal, your manuscript was reviewed by members of the editorial board and by several independent reviewers. The reviewers appreciated the attention to an important topic. Based on the reviews, we are likely to accept this manuscript for publication, providing that you modify the manuscript according to the few remaining recommendations by the reviewers (minor revision).

Sincerely,

Jens H. Kuhn

Associate Editor

PLOS Pathogens

Carolina Lopez

Section Editor

PLOS Pathogens

Kasturi Haldar

Editor-in-Chief

PLOS Pathogens

orcid.org/0000-0001-5065-158X

Michael Malim

Editor-in-Chief

PLOS Pathogens

orcid.org/0000-0002-7699-2064

Reviewer Comments (if any, and for reference):

Reviewer's Responses to Questions

**Part I - Summary**

Reviewer #1: In this resubmission, Zhang et. al. have made several important additions to their study, which provides evidence that GP5/M palmitoylation is important for producing infectious PRRS virus particles. I appreciate the detailed point by point responses to the critiques, and providing revisions, clarifications, and additional experiments to the manuscript where appropriate.

Reviewer #2: The submission is improved. New results in Fig 5 support a role for palmitoylation in particle assembly or secretion. A revised and clarified presentation of findings in Fig 9 confirm that that viruses containing under-palmitoylated GP5 have similar target-cell infection capacities. The inferences are that palmitoylation operates in virus-producing cells at the level of virus assembly, that some select under-palmitoylation can be tolerated at assembly, and that those under-palmitoylated virus particles that can be assembled and secreted will operate similarly to natural viruses during target cell entry.

Reviewer #3: Reviewed version of a paper.

**Part II – Major Issues: Key Experiments Required for Acceptance**

Reviewer #1: None

Reviewer #2: This reviewer still has a few questions.

1. Fig. 5: Since the vGP5 122 + 138, vM 99 and vM 102 are so prominent in several experiments (shown in Figs 4, 8, 9), why are they not evaluated and included in the Fig. 5 results?

2. The new Fig. 5 seems suboptimally placed into the manuscript. After producing mutant genomic cDNAs, the very first test would seem to be the one whose results are shown in Fig. 5. Then, with the results, the next questions could focus on the characteristics of those few recombinant cys-to-ser viruses that were produced and secreted; i.e., their replication kinetics (Fig 4), virion proteins (Fig 8), entry into target cells (Fig 9) and so on.

3. This reviewer still has difficult with summary statements. The statement at the end of the abstract refers to “clustering of GP5/M dimers at Golgi membranes” and it is not clear how the findings reported in the body of the abstract support this particular statement. Other summary statements are hard to discern, for example, lines 47-50 of the author summary; “acylation has no effect on dimerization of GP5 with M, its transport to budding sites, and incorporation into virus particles. We therefore propose that the fatty acids are required for the budding process.” It would help to clarify how observation of under-palmitoylated GP5 in virus particles lead one to the stated proposal.

Reviewer #3: I agree with the corrections made by the authors.

**Part III – Minor Issues: Editorial and Data Presentation Modifications**

Reviewer #1: 1. I appreciate the addition of Fig. 1A, and it’s fascinating to see this analysis of sequence variation between isolates. However, it’s difficult to discern which dot in the graph each cysteine is referring to given how scrunched together the data points are. To better distinguish the cysteines, could the authors simply mark those data points in a different color?

2. The addition of the PAM experiment (Fig. 2B) provides important data for a natural host cell target. However, I’m struggling with the M-protein blot. It doesn’t appear as though a -HA sample was loaded given the lack of spacing between the M-specific bands. Can the authors please comment, or perhaps include a new blot here that is more consistent with the others?

3. Lines 138-139: For added clarity, please denote which proteins ORF5 and ORF6 encode.

4. Line 176: Suggest replacing “confirming” to “consistent with”

Reviewer #2: (No Response)

Reviewer #3: I agree with the corrections made by the authors.

PLOS authors have the option to publish the peer review history of their article (what does this mean?). If published, this will include your full peer review and any attached files.

Reviewer #1: No

Reviewer #2: No

Reviewer #3: No

Figure Files:

Data Requirements:

Reproducibility:

References:

---

## [Editor Report · Decision Letter 2]

12 Apr 2021

Dear Dr. Veit,

We are pleased to inform you that your manuscript 'Palmitoylation of the envelope membrane proteins GP5 and M of porcine reproductive and respiratory syndrome virus is essential for virus growth' has been provisionally accepted for publication in PLOS Pathogens.

Best regards,

Jens H. Kuhn

Associate Editor

PLOS Pathogens

Carolina Lopez

Section Editor

PLOS Pathogens

Kasturi Haldar

Editor-in-Chief

PLOS Pathogens

orcid.org/0000-0001-5065-158X

Michael Malim

Editor-in-Chief

PLOS Pathogens

orcid.org/0000-0002-7699-2064
---

## [Editor Report · Acceptance letter]

20 Apr 2021

Dear Dr. Veit,

We are delighted to inform you that your manuscript, "Palmitoylation of the envelope membrane proteins GP5 and M of porcine reproductive and respiratory syndrome virus is essential for virus growth," has been formally accepted for publication in PLOS Pathogens.

Best regards,

Kasturi Haldar

Editor-in-Chief

PLOS Pathogens

orcid.org/0000-0001-5065-158X

Michael Malim

Editor-in-Chief

PLOS Pathogens

orcid.org/0000-0002-7699-2064